# Causal Fairness for Outcome Control

**Drago Plecko** and **Elias Bareinboim**
Department of Computer Science
Columbia University
dp3144@columbia.edu, eb@cs.columbia.edu

## Abstract

As society transitions towards an AI-based decision-making infrastructure, an ever-increasing number of decisions once under control of humans are now delegated to automated systems. Even though such developments make various parts of society more efficient, a large body of evidence suggests that a great deal of care needs to be taken to make such automated decision-making systems fair and equitable, namely, taking into account sensitive attributes such as gender, race, and religion. In this paper, we study a specific decision-making task called *outcome control* in which an automated system aims to optimize an outcome variable $Y$ while being fair and equitable. The interest in such a setting ranges from interventions related to criminal justice and welfare, all the way to clinical decision-making and public health. In this paper, we first analyze through causal lenses the notion of *benefit*, which captures how much a specific individual would benefit from a positive decision, counterfactually speaking, when contrasted with an alternative, negative one. We introduce the notion of benefit fairness, which can be seen as the minimal fairness requirement in decision-making, and develop an algorithm for satisfying it. We then note that the benefit itself may be influenced by the protected attribute, and propose causal tools which can be used to analyze this. Finally, if some of the variations of the protected attribute in the benefit are considered as discriminatory, the notion of benefit fairness may need to be strengthened, which leads us to articulating a notion of causal benefit fairness. Using this notion, we develop a new optimization procedure capable of maximizing $Y$ while ascertaining causal fairness in the decision process.

## 1 Introduction

Decision-making systems based on artificial intelligence and machine learning are being increasingly deployed in real-world settings where they have life-changing consequences on individuals and on society more broadly, including hiring decisions, university admissions, law enforcement, credit lending, health care access, and finance [19, 26, 6]. Issues of unfairness and discrimination are pervasive in those settings when decisions are being made by humans, and remain (or are potentially amplified) when decisions are made using machines with little transparency or accountability. Examples include reports on such biases in decision support systems for sentencing [1], face-detection [7], online advertising [37, 10], and authentication [35]. A large part of the underlying issue is that AI systems designed to make decisions are trained with data that contains various historical biases and past discriminatory decisions against certain protected groups, and such systems may potentially lead to an even more discriminatory process, unless they have a degree of fairness and transparency.

In this paper, we focus on the specific task of *outcome control*, characterized by a decision $D$ which precedes the outcome of interest $Y$. The setting of outcome control appears across a broad range of applications, from clinical decision-making [13] and public health [15], to criminal justice [23] and various welfare interventions [9]. In general, outcome control will cover settings in which an

37th Conference on Neural Information Processing Systems (NeurIPS 2023).

institution (such as a hospital or a social service) may attempt to maximize an outcome (such as survival or well-being) using a known control (such as surgery or a welfare intervention)[1]. Importantly, due to historical biases, certain demographic groups may differ in their distribution of covariates and therefore their benefit from treatment (e.g., from surgery or intervention) may be lower than expected, leading to a lower allocation of overall resources. We next discuss two lines of related literature.

Firstly, a large body of literature in reinforcement learning [36, 38] and policy learning [11, 33, 21, 18, 2] analyzes the task of optimal decision-making. Often, these works consider the conditional average treatment effect (CATE) that measures how much probabilistic gain there is from a positive versus a negative decision for a specific group of individuals when experimental data is available. Subsequent policy decisions are then based on the CATE, a quantity that will be important in our approach as well. The focus of this literature is often on developing efficient procedures with desirable statistical properties, and issues of fairness have not traditionally been explored in this context.

On the other hand, there is also a growing literature in fair machine learning, that includes various different settings. One can distinguish three specific and different tasks, namely (1) bias detection and quantification for currently deployed policies; (2) construction of fair predictions of an outcome; (3) construction of fair decision-making policies. Most of the work in fair ML falls under tasks (1) and (2), whereas our setting of outcome control falls under (3). Our work also falls under the growing literature that explores fairness through a causal lens [22, 20, 28, 44, 43, 41, 8, 32].

For concreteness, consider the causal diagram in Fig. 1 that represents the setting of outcome control, with $X$ the protected attribute, $Z$ a possibly multidimensional set of confounders, $W$ a set of mediators. Decision $D$ is based on the variables $X, Z$, and $W$, and the outcome $Y$ depends on all other variables in the model. In this setting, we also assume that the decision-maker is operating under budget constraints.

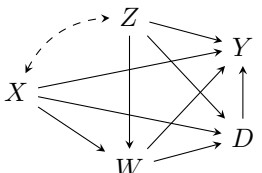

Figure 1: Standard Fairness Model (SFM) [31] extended for outcome control.

Previous work introduces a fairness definition that conditions on the potential outcomes of the decision $D$, written $Y_{d_0}, Y_{d_1}$, and ensures that the decision $D$ is independent of the protected attribute $X$ for any fixed value of $Y_{d_0}, Y_{d_1}$ [14]. Another related work, in a slightly different setting of risk assessment [9], proposes conditioning on the potential outcome under a negative decision, $Y_{d_0}$, and focuses on equalizing counterfactual error rates. Finally, the work [25] is also related to our setting, and proposes that fairness should be assessed with respect to three causal effects of interest: (i) total effect $X \rightarrow Y$; (ii) total effect $D \rightarrow Y$; and (iii) total effect $X \rightarrow D$. Our framework, however, integrates considerations of the above effects more coherently, and brings forward a new definition of fairness based on first principles.

However, the causal approaches mentioned above take a different perspective from the policy learning literature, in which policies are built based on the CATE of the decision $D$, written $\mathbb{E}[Y_{d_1} - Y_{d_0} \mid x, z, w]$, which we will refer to as *benefit*[2]. Focusing exclusively on the benefit, though, will provide no fairness guarantees apriori. In particular, as can be seen from Fig. 1, the protected attribute $X$ may influence the effect of $D$ on $Y$ in three very different ways: (i) along the direct pathway $X \rightarrow Y$; (ii) along the indirect pathway $X \rightarrow W \rightarrow Y$; (iii) along the spurious pathway $X \leftarrow\dashrightarrow Z \rightarrow Y$. Often, the decision-maker may view these causal effects differently, and may consider only some of them as discriminatory. Currently, no approach in the literature allows for a principled way of detecting and removing discrimination based on the notion of benefit, while accounting for different underlying causal mechanisms that may lead to disparities.

In light of the above, the goal of this paper is to analyze the foundations of outcome control from a causal perspective of the decision-maker. Specifically, we develop a causal-based decision-making framework for modeling fairness with the following contributions:

(i) We introduce Benefit Fairness (BF, Def. 2) to ensure that at equal levels of the benefit, the protected attribute $X$ is independent of the decision $D$. We then develop an algorithm for achieving BF (Alg. 1) and prove optimality guarantees (Thm. 2),

(ii) We develop Alg. 2 that determines which causal mechanisms from $X$ to the benefit (direct, indirect, spurious) explain the difference in the benefit between groups. The decision-maker can then decide which causal pathways are considered as discriminatory.

---

[1]Our setting is not related to the concept of outcome control in the procedural justice literature [24].

[2]We remark that the notion of benefit discussed in this work is different from that in [34].

(iii) We define the notion of causal benefit fairness (CBF, Def. 3), which models discriminatory pathways in a fine-grained way. We further develop an algorithm (Alg. 3) that ensures the removal of such discriminatory effects, and show some theoretical guarantees (Thm. 4).

## 1.1 Preliminaries

We use the language of structural causal models (SCMs) as our basic semantic framework [30]. A structural causal model (SCM) is a tuple $\mathcal{M} := \langle V, U, \mathcal{F}, P(u) \rangle$, where $V, U$ are sets of endogenous (observable) and exogenous (latent) variables respectively, $\mathcal{F}$ is a set of functions $f_{V_i}$, one for each $V_i \in V$, where $V_i \leftarrow f_{V_i}(\mathrm{pa}(V_i), U_{V_i})$ for some $\mathrm{pa}(V_i) \subseteq V$ and $U_{V_i} \subseteq U$. $P(u)$ is a strictly positive probability measure over $U$. Each SCM $\mathcal{M}$ is associated to a causal diagram $\mathcal{G}$ over the node set $V$, where $V_i \rightarrow V_j$ if $V_i$ is an argument of $f_{V_j}$, and $V_i \leftarrow\!\dashrightarrow V_j$ if the corresponding $U_{V_i}, U_{V_j}$ are not independent. Throughout, our discussion will be based on the specific causal diagram known as the standard fairness model (SFM, see Fig. 1). Further, an instantiation of the exogenous variables $U = u$ is called a *unit*. By $Y_x(u)$ we denote the potential response of $Y$ when setting $X = x$ for the unit $u$, which is the solution for $Y(u)$ to the set of equations obtained by evaluating the unit $u$ in the submodel $\mathcal{M}_x$, in which all equations in $\mathcal{F}$ associated with $X$ are replaced by $X = x$.

# 2 Foundations of Outcome Control

In the setting of outcome control, we are interested in the following decision-making task:

**Definition 1** (Decision-Making Optimization). *Let $\mathcal{M}$ be an SCM compatible with the SFM. We define the optimal decision problem as finding the (possibly stochastic) solution to the following optimization problem given a fixed budget $b$:*

$$D^* = \operatorname*{arg\,max}_{D(x,z,w)} \quad \mathbb{E}[Y_D] \tag{1}$$

$$\textit{subject to} \quad P(d) \leq b. \tag{2}$$

The budget $b$ constraint is relevant for scenarios when resources are scarce, in which not all patients possibly requiring treatment can be given treatment. In such a setting, the goal is to treat patients who are most likely to benefit from the treatment, as formalized later in the text. We next discuss two different perspectives on solving the above problem. First, we discuss the problem solution under perfect knowledge, assuming that the underlying SCM and the unobserved variables are available to us (we call this the oracle's perspective). Then, we move on to solving the problem from the point of view of the decision-maker, who only has access to the observed variables in the model and a dataset generated from the true SCM.

## 2.1 Oracle's Perspective

The following example, which will be used throughout the paper, is accompanied by a vignette that performs inference using finite sample data for the different computations described in the sequel. We introduce the example by illustrating the intuition of outcome control through the perspective of an all-knowing oracle:

**Example** (Cancer Surgery - continued). *A clinical team has access to information about the sex of cancer patients ($X = x_0$ male, $X = x_1$ female) and their degree of illness severity determined from tissue biopsy ($W \in [0, 1]$). They wish to optimize the 2-year survival of each patient ($Y$), and the decision $D = 1$ indicates whether to perform surgery. The following SCM $\mathcal{M}^*$ describes the data-generating mechanisms (unknown to the team):*

$$\mathcal{F}^*, P^*(U): \begin{cases} X \leftarrow U_X & (3) \\[2mm] W \leftarrow \begin{cases} \sqrt{U_W} \text{ if } X = x_0, \\ 1 - \sqrt{1 - U_W} \text{ if } X = x_1 \end{cases} & (4) \\[4mm] D \leftarrow f_D(X, W) & (5) \\[2mm] Y \leftarrow \mathbb{1}(U_Y + \frac{1}{3}WD - \frac{1}{5}W > 0.5). & (6) \\[2mm] U_X \in \{0, 1\}, P(U_X = 1) = 0.5, & (7) \\[2mm] U_W, U_Y \sim \textit{Unif}[0, 1], & (8) \end{cases}$$

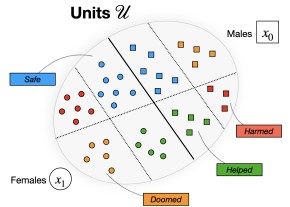

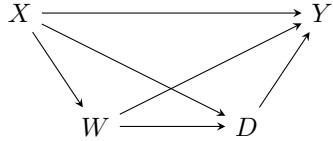

(a) Units in the oracle example.

(b) Causal diagram for the running example.

Figure 2: (a) Units in the Oracle example; (b) Standard Fairness Model (SFM) for the cancer surgery example, corresponding to $W = \{W\}$ and $Z = \emptyset$ in the SFM in Fig. 1.

*where the $f_D$ mechanism is constructed by the team.*

*The clinical team has access to an oracle that is capable of predicting the future perfectly. In particular, the oracle tells the team how each individual would respond to surgery. That is, for each unit $U = u$ (of the 500 units), the oracle returns the values of*

$$Y_{d_0}(u), Y_{d_1}(u). \tag{9}$$

*Having access to this information, the clinicians quickly realize how to use their resources. In particular, they notice that for units for whom $(Y_{d_0}(u), Y_{d_1}(u))$ equals $(0,0)$ or $(1,1)$, there is no effect of surgery, since they will (or will not) survive regardless of the decision. They also notice that surgery is harmful for individuals for whom $(Y_{d_0}(u), Y_{d_1}(u)) = (1,0)$. These individuals would not survive if given surgery, but would survive otherwise. Therefore, they ultimately decide to treat 100 individuals who satisfy*

$$(Y_{d_0}(u), Y_{d_1}(u)) = (0,1), \tag{10}$$

*since these individuals are precisely those whose death can be prevented by surgery. They learn there are 100 males and 100 females in the $(0,1)$-group, and thus, to be fair with respect to sex, they decide to treat 50 males and 50 females.* □

The space of units corresponding to the above example is represented in Fig. 2a. The groups described by different values of $Y_{d_0}(u), Y_{d_1}(u)$ in the example are known as *canonical types* [3] or *principal strata* [12]. Two groups cannot be influenced by the treatment decision (which will be called "Safe" and "Doomed", see Fig. 2a). The third group represents those who are harmed by treatment (called "Harmed"). In fact, the decision to perform surgery for this subset of individuals is harmful. Finally, the last group represents exactly those for whom the surgery is life-saving, which is the main goal of the clinicians (this group is called "Helped").

This example illustrates how, in presence of perfect knowledge, the team can allocate resources efficiently. In particular, the consideration of fairness comes into play when deciding which of the individuals corresponding to the $(0,1)$ principal stratum will be treated. Since the number of males and females in this group is equal, the team decides that half of those treated should be female. The approach described above can be seen as appealing in many applications unrelated to to the medical setting, and motivated the definition of principal fairness [14]. As we show next, however, this viewpoint is often incompatible with the decision-maker's perspective.

## 2.2 Decision-Maker's Perspective

We next discuss the policy construction from the perspective of the decision-maker:

**Example** (Cancer Surgery - continued)**.** *The team of clinicians constructs the causal diagram associated with the decision-making process, shown in Fig. 2b. Using data from their electronic health records (EHR), they estimate the benefit of the treatment based on $x, w$:*

$$\mathbb{E}[Y_{d_1} - Y_{d_0} \mid w, x_1] = \mathbb{E}[Y_{d_1} - Y_{d_0} \mid w, x_0] = \frac{w}{3}. \tag{11}$$

*In words, at each level of illness severity $W = w$, the proportion of patients who benefit from the surgery is the same, regardless of sex. In light of this information, the clinicians decide to construct*

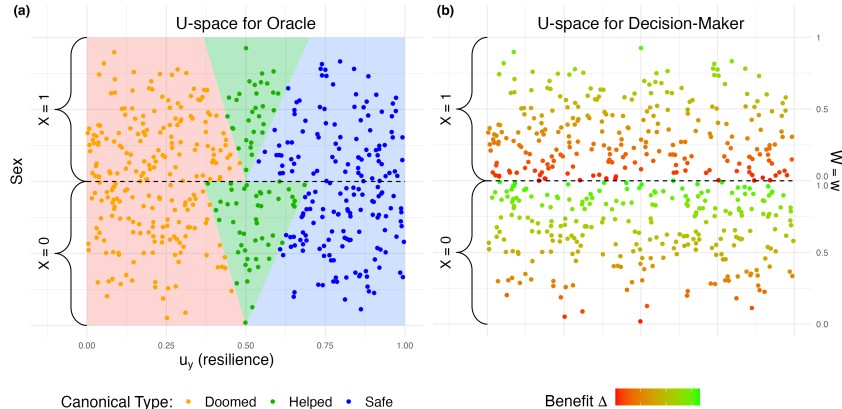

Figure 3: Difference in perspective between perfect knowledge of an oracle (left) and imperfect knowledge of a decision-maker in a practical application (right).

*the decision policy $f_D$ such that $f_D(W) = \mathbb{1}(W > \frac{1}{2})$. In words, if a patient's illness severity $W$ is above $\frac{1}{2}$, the patient will receive treatment.*

*After implementing the policy and waiting for the 2-year follow-up period, clinicians estimate the probabilities of treatment within the stratum of those helped by surgery, and compute that*

$$P(d \mid y_{d_0} = 0, y_{d_1} = 1, x_0) = \frac{7}{8}, \tag{12}$$

$$P(d \mid y_{d_0} = 0, y_{d_1} = 1, x_1) = \frac{1}{2}, \tag{13}$$

*indicating that the allocation of the decision is not independent of sex. That is, within the group of those who are helped by surgery, males are more likely to be selected for treatment than females.* $\square$

The decision-making policy introduced by the clinicians, somewhat counter-intuitively, does not allocate the treatment equally within the principal stratum of those who are helped, even though at each level of illness severity, the proportion of patients who benefit from the treatment is equal between the sexes. What is the issue at hand here?

To answer this question, the perspective under perfect knowledge is shown in Fig. 3a. In particular, on the horizontal axis the noise variable $u_y$ is available, which summarizes the patients' unobserved resilience. Together with the value of illness severity (on the vertical axis) and the knowledge of the structural causal model, we can perfectly pick apart the different groups (i.e., principal strata) according to their potential outcomes (groups are indicated by color). In this case, it is clear that our policy should treat patients in the green area since those are the ones who benefit from treatment.

In Fig. 3b, however, we see the perspective of the decision-makers under imperfect knowledge. Firstly, the decision-makers have no knowledge about the values on the horizontal axis, since this represents variables that are outside their model. From their point of view, the key quantity of interest is the conditional average treatment effect (CATE), which we call *benefit* in this context[3], defined as

$$\Delta(x, z, w) = P(y_{d_1} \mid x, z, w) - P(y_{d_0} \mid x, z, w). \tag{14}$$

Importantly, in our setting, the benefit $\Delta$ can be uniquely computed (identified) from observational data, implying that we may use the notion of benefit for practical data analyses:

**Proposition 1** (Benefit Identifiability). *Let $\mathcal{M}$ be an SCM compatible with the SFM in Fig. 1. Then the benefit $\Delta(x, z, w)$ is identifiable from observational data and equals:*

$$\Delta(x, z, w) = P(y \mid d_1, x, z, w) - P(y \mid d_0, x, z, w) \tag{15}$$

*for any choice of $x, z, w$.*

---

[3]In practice, however, clinical decision-making may not be based on the actual value of the benefit $\Delta$ but a biased, imperfect version of it, corresponding to the intuition of the clinician.

---

**Algorithm 1** Decision-Making with Benefit Fairness

---

• **Inputs:** Distribution $P(V)$, Budget $b$
1: Compute $\Delta(x, z, w) = \mathbb{E}[Y_{d_1} - Y_{d_0} \mid x, z, w]$ for all $(x, z, w)$.
2: If $P(\Delta > 0) \leq b$, set $D^* = \mathbb{1}(\Delta(x, z, w) > 0)$ and **RETURN**$(D^*)$.
3: Find $\delta_b > 0$ such that

$$P(\Delta > \delta_b) \leq b, P(\Delta \geq \delta_b) > b. \tag{17}$$

4: Otherwise, define

$$\mathcal{I} := \{(x, z, w) : \Delta(x, z, w) > \delta_b\}, \tag{18}$$
$$\mathcal{B} := \{(x, z, w) : \Delta(x, z, w) = \delta_b\} \tag{19}$$

5: Construct the policy $D^*$ such that:

$$D^* := \begin{cases} 1 \text{ for } (x, z, w) \in \mathcal{I}, \\ 1 \text{ with prob. } \frac{b - P(\mathcal{I})}{P(\mathcal{B})} \text{ for } (x_0, z, w) \in \mathcal{B}, \\ 1 \text{ with prob. } \frac{b - P(\mathcal{I})}{P(\mathcal{B})} \text{ for } (x_1, z, w) \in \mathcal{B}. \end{cases} \tag{20}$$

---

The benefit is simply the increase in survival associated with treatment. After computing the benefit (shown in Eq. 11), the decision-makers visualize the male and female groups (lighter color indicates a larger increase in survival associated with surgery) according to it. It is visible from the figure that the estimated benefit from surgery is higher for the $x_0$ group than for $x_1$. Therefore, to the best of their knowledge, the decision-makers decide to treat more patients from the $x_0$ group.

The example illustrates why the oracle's perspective may be misleading for the decision-makers. The clinicians can never determine exactly which patients belong to the group

$$Y_{d_0}(u) = 0, Y_{d_1}(u) = 1,$$

that is, who benefits from the treatment. Instead, they have to rely on illness severity ($W$) as a proxy for treatment benefit ($\Delta$). In other words, our understanding of treatment benefit will always be probabilistic, and we need to account for this when considering the decision-maker's task. A further discussion on the relation to principal fairness [14] is given in Appendix A.

## 2.3 Benefit Fairness

To remedy the above issue, we propose an alternative definition, which takes the viewpoint of the decision-maker:

**Definition 2** (Benefit Fairness). *Let $\Delta$ denote the benefit of treatment in Eq. 14 and let $\delta$ be a fixed value of $\Delta$. We say that the pair $(Y, D)$ satisfies benefit fairness (BF, for short) if*

$$P(d \mid \Delta = \delta, x_0) = P(d \mid \Delta = \delta, x_1) \ \ \forall \delta. \tag{16}$$

The above definition pertains to the population-level setting, but without oracle knowledge. In this setting, the objective function in Eq. 1 is expected to be lower than in the oracle case. Further, in finite samples, the constraint from Def. 2 is expected to hold approximately. Notice that BF takes the perspective of the decision-maker who only has access to the unit's attributes $(x, z, w)$, as opposed to the exogenous $U = u^4$. In particular, the benefit $\Delta(x, z, w)$ is *estimable* from the data, i.e., attainable by the decision-maker. BF then requires that at each level of the benefit, $\Delta(x, z, w) = \delta$, the rate of the decision does not depend on the protected attribute. The benefit $\Delta$ is closely related to the canonical types discussed earlier, namely

$$\Delta(x, z, w) = c(x, z, w) - d(x, z, w). \tag{21}$$

The values of $c, d$ in Eq. 21 are covariate-specific, and indicate the proportions of patients helped and harmed by the treatment, respectively, among all patients coinciding with covariate values $x, z, w$ (i.e.,

---

[4]Formally, having access to the exogenous instantiation of $U = u$ implies knowing which principal stratum from Fig. 2a the unit belongs to, since $U = u$ determines all of the variations of the model.

---

**Algorithm 2** Benefit Fairness Causal Explanation

---

- **Inputs:** Distribution $P(V)$, Benefit $\Delta(x, z, w)$, Decision policy $D$
1: Compute the causal decomposition of the resource allocation disparity into its direct, indirect, and spurious contributions [44, 31]

$$P(d \mid x_1) - P(d \mid x_0) = \text{DE} + \text{IE} + \text{SE}. \tag{22}$$

2: Compare the distributions $P(\Delta \mid x_1)$ and $P(\Delta \mid x_0)$.
3: Compute the causal decomposition of the benefit disparity

$$\mathbb{E}(\Delta \mid x_1) - \mathbb{E}(\Delta \mid x_0) = \text{DE} + \text{IE} + \text{SE}. \tag{23}$$

4: Compute the counterfactual distribution $P(\Delta_C \mid x)$ for specific interventions $C$ that remove the direct, indirect, or total effect of X on the benefit $\Delta$.

---

all $u \in U$ s.t. $(X, Z, W)(u) = (x, z, w)$). The proof of the claim in Eq. 21 is given in Appendix B. Using this connection of canonical types and the notion of benefit, we can formulate a solution to the problem in Def. 1, given in Alg. 1. In particular, Alg. 1 takes as input the observational distribution $P(V)$, but its adaptation to inference from finite samples follows easily. In Step 2, we check whether we are operating under resource scarcity, and if not, the optimal policy simply treats everyone who stands to benefit from the treatment. Otherwise, we find the $\delta_b > 0$ which uses the entire budget (Step 3) and separate the interior $\mathcal{I}$ of those with the highest benefit (all of whom are treated), and the boundary $\mathcal{B}$ (those who are to be randomized) in Step 4. The budget remaining to be spent on the boundary is $b - P(\mathcal{I})$, and thus individuals on the boundary are treated with probability $\frac{b - P(\mathcal{I})}{P(\mathcal{B})}$. Importantly, male and female groups are treated separately in this random selection process (Eq. 20), reflecting BF. The BF criterion can be seen as a minimal fairness requirement in decision-making, and is often aligned with maximizing utility, as seen from the following theorem:

**Theorem 2** (Alg. 1 Optimality). *Among all feasible policies $D$ for the optimization problem in Eqs. 1-2, the result of Alg. 1 is optimal and satisfies benefit fairness.*

A key extension we discuss next relates to the cases in which the benefit itself may be deemed as discriminatory towards a protected group.

## 3 Fairness of the Benefit

As discussed above, benefit fairness guarantees that at each fixed level of the benefit $\Delta = \delta$, the protected attribute plays no role in the treatment assignment. However, benefit fairness does not guarantee that treatment probability is equal between groups, i.e., that $P(d \mid x_1) = P(d \mid x_0)$. In fact, benefit fairness implies equal treatment allocation in cases where the distribution of the benefit $\Delta$ is equal across groups (shown formally in Appendix F), which may not always be the case:

**Example** (Cancer Surgery - continued). *After applying benefit fairness and implementing the optimal policy $D^* = \mathbb{1}\left(W > \frac{1}{2}\right)$, the clinicians compute that $P(d \mid x_1) - P(d \mid x_0) = -50\%$, that is, females are 50% less likely to be treated than males.* $\square$

In our example benefit fairness results in a disparity in resource allocation. Whenever this is the case, it implies that the benefit $\Delta$ differs between groups. In Alg. 2 we describe a formal procedure that helps the decision-maker to obtain a causal understanding of why that is, i.e., which underlying causal mechanisms (direct, indirect, spurious) lead to the difference in the benefit. In Appendix G we discuss in detail how the terms appearing in the decompositions in Eqs. 22 and 23 are defined. The key subtlety here is how to define counterfactuals with respect to the random variable $\Delta$. In the appendix we define this formally and show that the variable $\Delta$ can be considered as an auxillary variable in the structural causal model. Furthermore, we show that the notions of direct, indirect, and spurious effects are identifiable under the SFM in Fig. 1 and provide the identification expressions for them, allowing the data analyst to compute the decompositions in Alg. 2 in practice (see Appendix G). We next ground the idea behind Alg. 2 in our example:

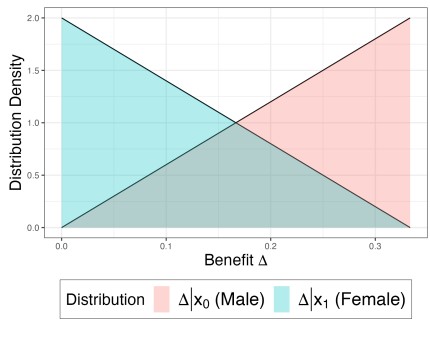
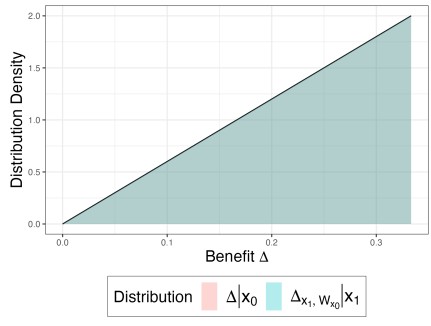

(a) Density of benefit $\Delta$ by group.

(b) Density of counterfactual benefit $\Delta_C$.

Figure 4: Elements of analytical tools from Alg. 2 in Ex. 3.

**Example** (Decomposing the disparity). *Following Alg. 2, the clinicians first decompose the observed disparities into their direct, indirect, and spurious components:*

$$P(d \mid x_1) - P(d \mid x_0) = \underbrace{0\%}_{DE} + \underbrace{-50\%}_{IE} + \underbrace{0\%}_{SE}, \tag{24}$$

$$\mathbb{E}(\Delta \mid x_1) - \mathbb{E}(\Delta \mid x_0) = \underbrace{0}_{DE} + \underbrace{-\frac{1}{9}}_{IE} + \underbrace{0}_{SE}, \tag{25}$$

*showing that the difference between groups is entirely explained by the levels of illness severity, that is, male patients are on average more severely ill than female patients (see Fig. 4a). Direct and spurious effects, in this example, do not explain the difference in benefit between the groups.*

*Based on these findings, the clinicians realize that the main driver of the disparity in the benefit $\Delta$ is the indirect effect. Thus, they decide to compute the distribution of the benefit $P(\Delta_{x_1, W_{x_0}} \mid x_1)$, which corresponds to the distribution of the benefit had $X$ been equal to $x_0$ along the indirect effect. The comparison of this distribution, with the distribution $P(\Delta \mid x_0)$ is shown in Fig. 4b, indicating that the two distributions are in fact equal.* □

In the above example, the difference between groups is driven by the indirect effect, although generally, the situation may be more complex, with a combination of effects driving the disparity. Still, the tools of Alg. 2 equip the reader for analyzing such more complex cases. The key takeaway here is that the first step in analyzing a disparity in treatment allocation is to obtain a *causal understanding* of why the benefit differs between groups. Based on this understanding, the decision-maker may decide that the benefit $\Delta$ is unfair, which is what we discuss next.

## 3.1 Controlling the Gap

**A causal approach.** The first approach for controlling the gap in resource allocation takes a counterfactual perspective. We first define what it means for the benefit $\Delta$ to be causally fair:

**Definition 3** (Causal Benefit Fairness). *Suppose $\mathcal{C} = (C_0, C_1)$ describes a causal pathway from $X$ to $Y$ which is deemed unfair, with $C_0, C_1$ representing possibly counterfactual interventions. The pair $(Y, D)$ satisfies counterfactual benefit fairness (CBF) if*

$$\mathbb{E}(y_{C_1, d_1} - y_{C_1, d_0} \mid x, z, w) = \mathbb{E}(y_{C_0, d_1} - y_{C_0, d_0} \mid x, z, w) \, \forall x, z, w \tag{26}$$

$$P(d \mid \Delta, x_0) = P(d \mid \Delta, x_1). \tag{27}$$

To account for discrimination along a specific causal pathway (after using Alg. 2), the decision-maker needs to compute an adjusted version of the benefit $\Delta$, such that the protected attribute has no effect along the intended causal pathway $\mathcal{C}$. For instance, $\mathcal{C} = (\{x_0\}, \{x_1\})$ describes the total causal effect, whereas $\mathcal{C} = (\{x_0\}, \{x_1, W_{x_0}\})$ describes the direct effect. In words, CBF requires that treatment benefit $\Delta$ should not depend on the effect of $X$ on $Y$ along the causal pathway $\mathcal{C}$, and this condition is covariate-specific, i.e., holds for any choice of covariates $x, z, w$. Additionally, the decision policy $D$ should satisfy BF, meaning that at each degree of benefit $\Delta = \delta$, the protected attribute plays no

role in deciding whether the individual is treated or not. This statement is for fixed value of $\Delta = \delta$, and possibly considers individuals with different values of $x, z, w$. CBF can be satisfied using Alg. 3. In Step 1, the factual benefit values $\Delta$, together with the adjusted, counterfactual benefit values $\Delta_C$ (that satisfy Def. 3) are computed. Then, $\delta_{CF}$ is chosen to match the budget $b$, and all patients with a counterfactual benefit above $\delta_{CF}$ are treated[5], as demonstrated in the following example:

**Example** (Cancer Surgery - Counterfactual Approach). *The clinicians realize that the difference in illness severity comes from the fact that female patients are subject to regular screening tests, and are therefore diagnosed earlier. The clinicians want to compute the adjusted benefit, by computing the counterfactual values of the benefit $\Delta_{x_1, W_{x_0}}(u)$ for all $u$ such that $X(u) = x_1$. For the computation, they assume that the relative order of the illness severity for females in the counterfactual world would have stayed the same (which holds true in the underlying SCM). Therefore, they compute that*

$$\Delta_{x_1, W_{x_0}}(u) = \frac{1}{3}\sqrt{1 - (1 - W(u))^2}, \tag{28}$$

*for each unit $u$ with $X(u) = x_1$. After applying Alg. 1 with the counterfactual benefit values $\Delta_C$, the resulting policy $D^{CF} = \mathbb{1}(\Delta_C > \frac{1}{4})$ has a resource allocation disparity of $0$.* $\qquad\square$

The above example illustrates the core of the causal counterfactual approach to discrimination removal. BF was not appropriate in itself, since the clinicians are aware that the benefit of the treatment depends on sex in a way they deemed unfair. Therefore, to solve the problem, they first remove the undesired effect from the benefit $\Delta$, by computing the counterfactual benefit $\Delta_C$. After this, they apply Alg. 3 with the counterfactual method (CF) to construct a fair decision policy. Notably, the causal approach to reducing the disparity relies on the counterfactual values of the benefit $\Delta_C(u)$, as opposed to the factual benefit values $\Delta(u)$. As it turns out, measuring (or removing) the covariate-specific direct effect of $X$ on the benefit $\Delta$ is relatively simple in general, while the indirect effect may be more challenging to handle:

**Remark 3** (Covariate-Specific Direct Effect of $X$ on Benefit $\Delta$ is Computable). *Under the assumptions of the SFM, the potential outcome $\Delta_{x_1, W_{x_0}}(u)$ is identifiable for any unit $u$ with $X(u) = x_0$ for which the attributes $Z(u) = z, W(u) = w$ are observed. However, the same is not true for the indirect effect. While the counterfactual distribution of the benefit when the indirect effect is manipulated, written $P(\Delta_{x_0, W_{x_1}} \mid x_0)$, is identifiable under the SFM, the covariate-level values $\Delta_{x_0, W_{x_1}}(x_0, z, w)$ are not identifiable without further assumptions.*

**A utilitarian/factual approach.** An alternative, utilitarian (or factual) approach to reduce the disparity in resource allocation uses the factual benefit $\Delta(u)$, instead of the counterfactual benefit $\Delta_C(u)$ used in the causal approach. This approach is also described in Alg. 3, with the utilitarian (UT) method. Firstly, in Step 5, the counterfactual values $\Delta_C$ are used to compute the disparity that would arise from the optimal policy in the hypothetical, counterfactual world:

$$M := |P(\Delta_C \geq \delta_{CF} \mid x_1) - P(\Delta_C \geq \delta_{CF} \mid x_0)|. \tag{29}$$

The idea then is to introduce different thresholds $\delta^{(x_0)}, \delta^{(x_1)}$ for $x_0$ and $x_1$ groups, such that they introduce a disparity of at most $M$. In Step 6 we check whether the optimal policy introduces a disparity bounded by $M$. If the disparity is larger than $M$ by an $\epsilon$, in Step 7 we determine how much slack the disadvantaged group requires, by finding thresholds $\delta^{(x_0)}, \delta^{(x_1)}$ that either treat everyone in the disadvantaged group, or achieve a disparity bounded by $M$. The counterfactual (CF) approach focused on the counterfactual benefit values $\Delta_C$ and used a single threshold. The utilitarian (UT) approach focuses on the factual benefit values $\Delta$, but uses different thresholds within groups. However, the utilitarian approach uses the counterfactual values $\Delta_C$ to determine the maximum allowed disparity. Alternatively, this disparity can be pre-specified, as shown in the following example:

**Example** (Cancer Surgery - Utilitarian Approach). *Due to regulatory purposes, clinicians decide that $M = 20\%$ is the maximum allowed disparity that can be introduced by the new policy $D$. Using Alg. 3, they construct $D^{UT}$ and find that for $\delta^{(x_0)} = 0.21, \delta^{(x_1)} = 0.12$,*

$$P(\Delta > \delta^{(x_0)} \mid x_0) \approx 60\%, P(\Delta > \delta^{(x_1)} \mid x_1) \approx 40\%, \tag{33}$$

*which yields $P(d^{UT}) \approx 50\%$, and $P(d^{UT} \mid x_1) - P(d^{UT} \mid x_0) \approx 20\%$, which is in line with the hospital resources and the maximum disparity allowed by the regulators.* $\qquad\square$

---

[5]In this section, for clarity of exposition we assume that distribution of the benefit admits a density, although the methods are easily adapted to the case when this does not hold.

---

**Algorithm 3** Causal Discrimination Removal for Outcome Control

---

• **Inputs:** Distribution $P(V)$, Budget $b$, Intervention $C$, Max. Disparity $M$, Method $\in \{CF, UT\}$
1: Compute $\Delta(x, z, w), \Delta_C(x, z, w)$ for all $(x, z, w)$.
2: If $P(\Delta > 0) \leq b$, set $D = \mathbb{1}(\Delta(x, z, w) > 0)$ and **RETURN**($D$).
3: Find $\delta_{CF} > 0$ such that

$$P(\Delta_C \geq \delta_{CF}) = b. \tag{30}$$

4: If Method is CF, set $D^{CF} = \mathbb{1}(\Delta_C(x, z, w) \geq \delta_{CF})$ and **RETURN**($D^{CF}$).
5: If $M$ not pre-specified, compute the disparity

$$M := P(\Delta_C \geq \delta_{CF} \mid x_1) - P(\Delta_C \geq \delta_{CF} \mid x_0). \tag{31}$$

6: Find $\delta_{UT}$ such that $P(\Delta \geq \delta_{UT}) = b$. If

$$|P(\Delta \geq \delta_{UT} \mid x_1) - P(\Delta \geq \delta_{UT} \mid x_0)| \leq M,$$

set $D^{UT} = \mathbb{1}(\Delta(x, z, w) \geq \delta_{UT})$ and **RETURN**($D^{UT}$).
7: Otherwise, suppose w.l.o.g. that $P(\Delta \geq \delta_b \mid x_1) - P(\Delta \geq \delta_b \mid x_0) = M + \epsilon$ for $\epsilon > 0$. Define $l := \frac{P(x_1)}{P(x_0)}$, and let $\delta_{lb}^{(x_0)}$ be such that

$$P(\Delta \geq \delta_{lb}^{(x_0)} \mid x_0) = P(\Delta \geq \delta_b \mid x_0) + \epsilon \frac{l}{1+l}.$$

Set $\delta^{(x_0)} = \max(\delta_{lb}^{(x_0)}, 0)$, and $\delta^{(x_1)}$ s.t. $P(\Delta \geq \delta^{(x_1)} \mid x_1) = \frac{b}{P(x_1)} - \frac{1}{l}P(\Delta \geq \delta^{(x_0)} \mid x_0)$.
8: Construct and **RETURN** the policy $D^{UT}$:

$$D^{UT} := \begin{cases} 1 \text{ for } (x_1, z, w) \text{ s.t. } \Delta(x_1, z, w) \geq \delta^{(x_1)}, \\ 1 \text{ for } (x_0, z, w) \text{ s.t. } \Delta(x_0, z, w) \geq \delta^{(x_0)}, \\ 0 \text{ otherwise.} \end{cases} \tag{32}$$

---

Finally, we describe the theoretical guarantees for the methods in Alg. 3 (proof given in Appendix C):

**Theorem 4** (Alg. 3 Guarantees). *The policy $D^{CF}$ is optimal among all policies with a budget $\leq b$ that in the counterfactual world described by intervention $C$. The policy $D^{UT}$ is optimal among all policies with a budget $\leq b$ that either introduce a bounded disparity in resource allocation $|P(d \mid x_1) - P(d \mid x_0)| \leq M$ or treat everyone with a positive benefit in the disadvantaged group.*

We remark that policies $D^{CF}$ and $D^{UT}$ do not necessarily treat the same individuals in general. In Appendix D, we discuss a formal condition called *counterfactual crossing* that ensures that $D^{CF}$ and $D^{UT}$ treat the same individuals, therefore explaining when the causal and utilitarian approaches are equivalent [29]. In Appendix E we provide an additional application of our outcome control framework to the problem of allocating respirators [5] in intensive care units (ICUs), using the MIMIC-IV dataset [17].

## 4 Conclusion

In this paper we developed causal tools for understanding fairness in the task of outcome control. We introduced the notion of benefit fairness (Def. 2), and developed a procedure for achieving it (Alg. 1). Further, we develop a procedure for determining which causal mechanisms (direct, indirect, spurious) explain the difference in benefit between groups (Alg. 2). Finally, we developed two approaches that allow the removal of discrimination from the decision process along undesired causal pathways (Alg. 3). The proposed framework was demonstrated through a hypothetical cancer surgery example (see vignette) and a real-world respirator allocation example (see Appendix E). We leave for future work the extensions of the methods to the setting of continuous decisions $D$, and the setting of performing decision-making under uncertainty or imperfect causal knowledge.

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

**Acknowledgements**   This research was supported in part by the NSF, ONR, AFOSR, DoE, Amazon, JP Morgan, and The Alfred P. Sloan Foundation.

# Supplementary Material for *Causal Fairness for Outcome Control*

The source code for reproducing all the experiments can be found in our code repository. Furthermore, the vignette accompanying the main text can be found here.

## A  Principal Fairness

We start with the definition of principal fairness:

**Definition 4** (Principal Fairness [14]). *Let $D$ be a decision that possibly affects the outcome $Y$. The pair $(Y, D)$ is said to satisfy principal fairness if*

$$P(d \mid y_{d_0}, y_{d_1}, x_1) = P(d \mid y_{d_0}, y_{d_1}, x_0), \tag{34}$$

*for each principal stratum $(y_{d_0}, y_{d_1})$, which can also be written as $D \perp\!\!\!\perp X \mid Y_{d_0}, Y_{d_1}$. Furthermore, define the principal fairness measure (PFM) as:*

$$PFM_{x_0, x_1}(d \mid y_{d_0}, y_{d_1}) = P(d \mid y_{d_0}, y_{d_1}, x_1) - P(d \mid y_{d_0}, y_{d_1}, x_0). \tag{35}$$

The above notion of principal fairness aims to capture the intuition described in oracle example in Sec. 2.1. However, unlike in the example, the definition needs to be evaluated under imperfect knowledge, when only the collected data is available[6]. An immediate cause for concern, in this context, is the joint appearance of the potential outcomes $Y_{d_0}, Y_{d_1}$ in the definition of principal fairness. As is well-known in the literature, the joint distribution of the potential outcomes $Y_{d_0}, Y_{d_1}$ is in general impossible to obtain, which leads to the lack of identifiability of the principal fairness criterion:

**Proposition 5** (Principal Fairness is Not Identifiable). *The Principal Fairness (PF) criterion from Eq. 34 is not identifiable from observational or experimental data.*

The implication of the proposition is that principal fairness, in general, cannot be evaluated, even if an unlimited amount of data was available. One way to see why PF is not identifiable is the following construction. Consider an SCM consisting of two binary variables $D, Y \in \{0, 1\}$ and the simple graph $D \to Y$. Suppose that we observe $P(d) = p_d$, and $P(y \mid d_1) = m_1$, $P(y \mid d_0) = m_0$ for some constants $p_d, m_1, m_0$ (additionally assume $m_0 \leq m_1$ w.l.o.g.). It is easy to show that these three values determine all of the observational and interventional distributions of the SCM. However, notice that for any $\lambda \in [0, 1 - m_1]$ the SCM given by

$$D \leftarrow U_D \tag{36}$$

$$Y \leftarrow \mathbb{1}(U_Y \in [0, m_0 - \lambda]) + D\mathbb{1}(U_Y \in [m_0 - \lambda, m_1]) + \tag{37}$$
$$(1 - D)\mathbb{1}(U_Y \in [m_1, m_1 + \lambda]),$$

$$U_Y \sim \text{Unif}[0, 1], U_D \sim \text{Bernoulli}(p_d), \tag{38}$$

satisfies $P(d) = p_d, P(y \mid d_1) = m_1$, and $P(y \mid d_0) = m_0$, but the joint distribution $P(y_{d_0} = 0, y_{d_1} = 1) = m_1 - m_0 + \lambda$ depends on the $\lambda$ parameter and is therefore non-identifiable.

### A.1  Monotonicity Assumption

To remedy the problem of non-identifiability of principal fairness, [14] proposes the monotonicity assumption:

**Definition 5** (Monotonicity). *We say that an outcome $Y$ satisfies monotonicity with respect to a decision $D$ if*

$$Y_{d_1}(u) \geq Y_{d_0}(u). \tag{39}$$

In words, monotonicity says that for every unit, the outcome with the positive decision ($D = 1$) would not be worse than with the negative decision ($D = 0$). We now demonstrate how monotonicity aids the identifiability of principal fairness.

---

[6]As implied by the definition of the SCM, we almost never have access to the unobserved sources of variation ($u$) that determine the identity of each unit.

**Proposition 6.** *Under the monotonicity assumption (Eq. 39), the principal fairness criterion is identifiable under the Standard Fairness Model (SFM).*

*Proof.* The main challenge in PF is to obtain the joint distribution $P(y_{d_0}, y_{d_1})$, which is non-identifiable in general. Under monotonicity, however, we have that

$$Y_{d_0}(u) = 0 \wedge Y_{d_1}(u) = 0 \iff Y_{d_1}(u) = 0, \tag{40}$$

$$Y_{d_0}(u) = 1 \wedge Y_{d_1}(u) = 1 \iff Y_{d_0}(u) = 1. \tag{41}$$

Therefore, it follows from monotonicity that

$$P(y_{d_0} = 1, y_{d_1} = 0) = 0, \tag{42}$$

$$P(y_{d_0} = 0, y_{d_1} = 0) = P(y_{d_1} = 0), \tag{43}$$

$$P(y_{d_0} = 1, y_{d_1} = 1) = P(y_{d_0} = 1), \tag{44}$$

$$P(y_{d_0} = 0, y_{d_1} = 1) = 1 - P(y_{d_1} = 0) - P(y_{d_0} = 1), \tag{45}$$

thereby identifying the joint distribution whenever the interventional distributions $P(y_{d_0}), P(y_{d_1})$ are identifiable. $\square$

In the cancer surgery example, the monotonicity assumption would require that the patients have strictly better survival outcomes when the surgery is performed, compared to when it is not. Given the known risks of surgical procedures, the assumption may be rightfully challenged in such a setting. In the sequel, we argue that the assumption of monotonicity is not really necessary, and often does not help the decision-maker, even if it holds true. To fix this issue, in the main text we discuss a relaxation of the PF criterion that suffers from neither of the above two problems but still captures the essential intuition that motivated PF.

# B    Canonical Types & Bounds

**Definition 6** (Canonical Types for Decision-Making). *Let $Y$ be the outcome of interest, and $D$ a binary decision. We then consider four canonical types of units:*

    *(i)  units $u$ such that $Y_{d_0}(u) = 1, Y_{d_1}(u) = 1$, called* **safe,**

    *(ii)  units $u$ such that $Y_{d_0}(u) = 1, Y_{d_1}(u) = 0$, called* **harmed,**

    *(iii)  units $u$ such that $Y_{d_0}(u) = 0, Y_{d_1}(u) = 1$, called* **helped,**

    *(iv)  units $u$ such that $Y_{d_0}(u) = 0, Y_{d_1}(u) = 0$, called* **doomed.**

In decision-making, the goal is to treat as many units who are helped by the treatment, and as few who are harmed by it. As we demonstrate next, the potential outcomes $Y_{d_0}(u), Y_{d_1}(u)$ depend precisely on the canonical types described above.

**Proposition 7** (Canonical Types Decomposition). *Let $\mathcal{M}$ be an SCM compatible with the SFM. Let $D$ be a binary decision that possibly affects the outcome $Y$. Denote by $(s, d, c, u)(x, z, w)$ the proportion of each of the canonical types from Def. 6, respectively, for a fixed set of covariates $(x, z, w)$. It then holds that*

$$P(y_{d_1} \mid x, z, w) = c(x, z, w) + s(x, z, w), \tag{46}$$

$$P(y_{d_0} \mid x, z, w) = d(x, z, w) + s(x, z, w). \tag{47}$$

*Therefore, we have that*

$$\Delta(x, z, w) := P(y_{d_1} \mid x, z, w) - P(y_{d_0} \mid x, z, w) \tag{48}$$

$$= c(x, z, w) - d(x, z, w). \tag{49}$$

*Proof.* Notice that we can write:

$$P(y_{d_1} \mid x, z, w) = P(y_{d_1} = 1, y_{d_0} = 1 \mid x, z, w) + P(y_{d_1} = 1, y_{d_0} = 0 \mid x, z, w) \tag{50}$$

$$= s(x, z, w) + c(x, z, w). \tag{51}$$

where the first line follows from the law of total probability, and the second by definition. Similarly, we have that

$$P(y_{d_0} \mid x, z, w) = P(y_{d_0} = 1, y_{d_1} = 1 \mid x, z, w) + P(y_{d_0} = 1, y_{d_1} = 0 \mid x, z, w) \tag{52}$$
$$= s(x, z, w) + d(x, z, w), \tag{53}$$

thereby completing the proof. $\qquad \square$

The proposition shows us that the degree of benefit $\Delta(x, z, w)$ captures exactly the difference between the proportion of those helped by the treatment, versus those who are harmed by it. From the point of view of the decision-maker, this is very valuable information since higher $\Delta(x, z, w)$ values indicate a higher utility of treating the group corresponding to covariates $(x, z, w)$. This insight can be used to prove Thm. 2, which states that the policy $D^*$ obtained by Alg. 1 is optimal:

*Proof.* Note that the objective in Eq. 1 can be written as:

$$\mathbb{E}[Y_D] = P(Y_D = 1) \tag{54}$$
$$= \sum_{x,z,w} P(Y_D = 1 \mid x, z, w) P(x, z, w) \tag{55}$$
$$= \sum_{x,z,w} \Big[ P(Y_{d_1} = 1, D = 1 \mid x, z, w) + P(Y_{d_0} = 1, D = 0 \mid x, z, w) \Big] P(x, z, w). \tag{56}$$

Eq. 55 follows from the law of total probability, and Eq. 56 from the consistency axiom. Now, note that $Y_{d_0}, Y_{d_1} \perp\!\!\!\perp D \mid X, Z, W$, from which it follows that

$$\mathbb{E}[Y_D] = \sum_{x,z,w} \Big[ P(y_{d_1} \mid x, z, w) P(D = 1 \mid x, z, w) + P(y_{d_0} \mid x, z, w) P(D = 0 \mid x, z, w) \Big] \tag{57}$$
$$* P(x, z, w).$$

By noting that $P(D = 0 \mid x, z, w) = 1 - P(D = 1 \mid x, z, w)$, we can rewrite the objective as

$$\sum_{x,z,w} \Big[ (s(x, z, w) + c(x, z, w)) P(d \mid x, z, w) \tag{58}$$
$$+ (s(x, z, w) + d(x, z, w))(1 - P(d \mid x, z, w)) \Big] P(x, z, w)$$
$$= \sum_{x,z,w} \Big[ s(x, z, w) + P(d \mid x, z, w)[c(x, z, w) - d(x, z, w)] \Big] P(x, z, w) \tag{59}$$
$$= P(y_{d_0} = 1, y_{d_1} = 1) + \sum_{x,z,w} P(d \mid x, z, w) P(x, z, w) \Delta(x, z, w). \tag{60}$$

Only the second term in Eq. 60 can be influenced by the decision-maker, and optimizing the term is subject to the budget constraint:

$$\sum_{x,z,w} P(d \mid x, z, w) P(x, z, w) \leq b. \tag{61}$$

Such an optimization problem is a simple linear programming exercise, for which the policy $D^*$ from Alg. 1 is a (possibly non-unique) optimal solution. $\qquad \square$

Finally, as the next proposition shows, the values of $P(y_{d_1} \mid x, z, w), P(y_{d_0} \mid x, z, w)$ can be used to bound the proportion of different canonical types:

**Proposition 8** (Canonical Types Bounds and Tightness). *Let $(s, d, c, u)(x, z, w)$ denote the proportion of each of the canonical types from Def. 6 for a fixed set of covariates $(x, z, w)$. Let $m_1(x, z, w) = P(y_{d_1} \mid x, z, w)$ and $m_0(x, z, w) = P(y_{d_0} \mid x, z, w)$ and suppose that $m_1(x, z, w) \geq m_0(x, z, w)$. We then have that (dropping $(x, z, w)$ from the notation):*

$$d \in [0, \min(m_0, 1 - m_1)], \tag{62}$$
$$c \in [m_1 - m_0, m_1]. \tag{63}$$

*In particular, the above bounds are tight, meaning that there exists an SCM $\mathcal{M}$, compatible with the observed data, that attains each of the values within the interval. Under monotonicity, the bounds collapse to single points, with $d = 0$ and $c = m_1 - m_0$.*

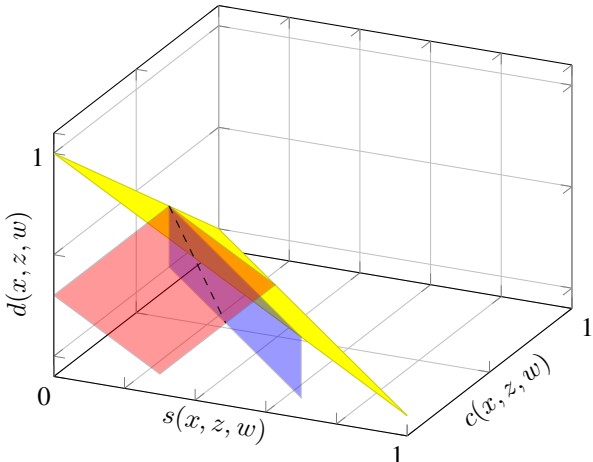

Figure 5: Canonical types solution space. The unit simplex is shown in yellow, the $s + c = m_1$ plane in blue, and the $s + d = m_0$ plane in red. The solution space for the possible values of $\big(s(x, z, w), c(x, z, w), d(x, z, w)\big)$ lies at the intersection of the red and blue planes, indicated by the dashed black line.

*Proof.* There are three linear relations that the values $s, d, c, u$ obey:

$$s + c = m_1, \tag{64}$$
$$s + d = m_0, \tag{65}$$
$$s + u + d + c = 1. \tag{66}$$

On top of this, we know that $s, d, c, u$ are all non-negative. Based on the linear relations, we know that the following parametrization of the vector $(s, d, c, u)$ holds

$$(s, d, c, u) = (m_0 - d, d, d + m_1 - m_0, 1 - m_1 - d), \tag{67}$$

which represents a line in the 3-dimensional space $(s, d, c)$. In particular, we know that the values of $(s, d, c)$ have to lie below the unit simplex in Fig. 5 (in yellow). In particular, the red and the blue planes represent the linear constraints from Eq. 64-65. The line parametrized in Eq. 67 lies at the intersection of the red and blue planes. Notice that $d \in [0, \min(m_0, 1 - m_1)]$ since each of the elements in Eq. 67 is positive. This bound on $d$ also implies that $c \in [m_1 - m_0, m_1]$. Finally, we need to construct an $f_Y$ mechanism that achieves any value within the bounds. To this end, define

$$f_Y(x, z, w, d, u_y) = \mathbb{1}(u_y \in [0, s]) + d * \mathbb{1}(u_y \in [s, s + c]) + \tag{68}$$
$$(1 - d) * \mathbb{1}(u_y \in [s + c, s + c + d]), \tag{69}$$
$$u_y \sim \text{Unif}[0, 1]. \tag{70}$$

which is both feasible and satisfies the proportion of canonical types to be $(s, d, c, u)$. $\qquad\square$

## C    Proof of Thm. 4

*Proof.* The first part of the theorem states the optimality of the $D^{CF}$ policy in the counterfactual world. Given that the policy uses the true benefit values from the counterfactual world, we apply the argument of Thm. 2 to prove its optimality.

We next prove the optimality of the $D^{UT}$ policy from Alg. 3. In Step 2 we check whether all individuals with a positive benefit can be treated. If yes, then the policy $D^{UT}$ is the overall optimal policy. If not, in Step 6 we check whether the overall optimal policy has a disparity bounded by $M$. If this is the case, $D^{UT}$ is the overall optimal policy for a budget $\leq b$, and cannot be strictly improved. For the remainder of the proof, we may suppose that $D^{UT}$ uses the entire budget $b$ (since we are operating under scarcity), and that $D^{UT}$ has introduces a disparity $\geq M$. We also assume that the benefit $\Delta$ admits a density, and that probability $P(\Delta \in [a, b] \mid x) > 0$ for any $[a, b] \subset [0, 1]$ and $x$.

Let $\delta^{(x_0)}, \delta^{(x_1)}$ be the two thresholds used by the $D^{UT}$ policy. Suppose that $\widetilde{D}^{UT}$ is a policy that has a higher expected utility and introduces a disparity bounded by $M$, or treats everyone in the disadvantaged group. Then there exists an alternative policy $\overline{D}^{UT}$ with a higher or equal utility that takes the form

$$\overline{D}^{UT} = \begin{cases} 1 \text{ if } \Delta(x_1, z, w) > \delta^{(x_1)'}, \\ 1 \text{ if } \Delta(x_0, z, w) > \delta^{(x_0)'}, \\ 0 \text{ otherwise.} \end{cases} \tag{71}$$

with $\delta^{(x_0)'}, \delta^{(x_1)'}$ non-negative (otherwise, the policy can be trivially improved). In words, for any policy $\overline{D}^{UT}$ there is a threshold based policy that is no worse. The policy $D^{UT}$ is also a threshold based policy. Now, if we had

$$\delta^{(x_1)'} < \delta^{(x_1)} \tag{72}$$

$$\delta^{(x_0)'} < \delta^{(x_0)} \tag{73}$$

it would mean policy $\overline{D}^{UT}$ is using a larger budget than $D^{UT}$. However, $D^{UT}$ uses a budget of $b$, making $\overline{D}^{UT}$ infeasible. Therefore, we must have that

$$\delta^{(x_1)'} < \delta^{(x_1)}, \delta^{(x_0)'} > \delta^{(x_0)} \text{ or} \tag{74}$$

$$\delta^{(x_1)'} > \delta^{(x_1)}, \delta^{(x_0)'} < \delta^{(x_0)}. \tag{75}$$

We first handle the case in Eq. 74. In this case, the policy $\overline{D}^{UT}$ introduces a larger disparity than $D^{UT}$. Since the disparity of $D^{UT}$ is at least $M$, the disparity of $\overline{D}^{UT}$ is strictly greater than $M$. Further, note that $\delta^{(x_0)'} > \delta^{(x_0)} \geq 0$, showing that $\overline{D}^{UT}$ does not treat all individuals with a positive benefit in the disadvantaged group. Combined with a disparity of $> M$, this makes the policy $\overline{D}^{UT}$ infeasible.

For the second case in Eq. 75, let $U(\delta_0, \delta_1)$ denote the utility of a threshold based policy:

$$U(\delta_0, \delta_1) = \mathbb{E}[\Delta \mathbb{1}(\Delta > \delta_0)\mathbb{1}(X = x_0)] + \mathbb{E}[\Delta \mathbb{1}(\Delta > \delta_1)\mathbb{1}(X = x_1)]. \tag{76}$$

Thus, we have that

$$U(\delta^{(x_0)}, \delta^{(x_1)}) - U(\delta^{(x_0)'}, \delta^{(x_1)'}) = \mathbb{E}[\Delta \mathbb{1}(\Delta \in [\delta^{(x_1)}, \delta^{(x_1)'}])\mathbb{1}(X = x_1)] \tag{77}$$

$$- \mathbb{E}[\Delta \mathbb{1}(\Delta \in [\delta^{(x_0)'}, \delta^{(x_0)}])\mathbb{1}(X = x_0)] \tag{78}$$

$$\geq \delta^{(x_1)} \mathbb{E}[\mathbb{1}(\Delta \in [\delta^{(x_1)}, \delta^{(x_1)'}])\mathbb{1}(X = x_1)] \tag{79}$$

$$- \delta^{(x_0)} \mathbb{E}[\mathbb{1}(\Delta \in [\delta^{(x_0)'}, \delta^{(x_0)}])\mathbb{1}(X = x_0)] \tag{80}$$

$$\geq \delta^{(x_0)} \big( \mathbb{E}[\mathbb{1}(\Delta \in [\delta^{(x_1)}, \delta^{(x_1)'}])\mathbb{1}(X = x_1)] \tag{81}$$

$$- \mathbb{E}[\mathbb{1}(\Delta \in [\delta^{(x_0)'}, \delta^{(x_0)}])\mathbb{1}(X = x_0)] \big) \tag{82}$$

$$= \delta^{(x_0)} \big( P(\Delta \in [\delta^{(x_1)}, \delta^{(x_1)'}], x_1) \tag{83}$$

$$- P(\Delta \in [\delta^{(x_0)'}, \delta^{(x_0)}], x_0) \big) \tag{84}$$

$$\geq 0, \tag{85}$$

where the last line follows from the fact that $\overline{D}^{UT}$ has a budget no higher than $D^{UT}$. Thus, this case also gives a contradiction.

Therefore, we conclude that policy $D^{UT}$ is optimal among all policies with a budget $\leq b$ that either introduce a bounded disparity in resource allocation $|P(d \mid x_1) - P(d \mid x_0)| \leq M$ or treat everyone with a positive benefit in the disadvantaged group. $\qquad\square$

## D   Equivalence of CF and UT Methods in Alg. 3

A natural question to ask is whether the two methods in Alg. 3 yield the same decision policy in terms of the individuals that are selected for treatment. To examine this issue, we first define the notion of counterfactual crossing:

**Definition 7** (Counterfactual crossing). *We say that two units of the population $u_1, u_2$ satisfy counterfactual crossing with respect to an intervention $C$ if*

 *(i) $u_1, u_2$ belong to the same protected group, $X(u_1) = X(u_2)$.*

 *(ii) unit $u_1$ has a higher factual benefit than $u_2$, $\Delta(u_1) > \Delta(u_2)$,*

 *(iii) unit $u_1$ has a lower counterfactual benefit than $u_2$ under the intervention $C$, $\Delta_C(u_1) < \Delta_C(u_2)$.*

In words, two units satisfy counterfactual crossing if $u_1$ has a higher benefit than $u_2$ in the factual world, while in the counterfactual world the benefit is larger for the unit $u_2$. Based on this notion, we can give a condition under which the causal and utilitarian approaches are equivalent:

**Proposition 9** (Causal and Utilitarian Equivalence). *Suppose that no two units of the population satisfy counterfactual crossing with respect to an intervention $C$, and suppose that the distribution of the benefit $\Delta$ admits a density. Then, the causal approach based on applying Alg. 1 with counterfactual benefit $\Delta_C$, and the utilitarian approach based on factual benefit $\Delta$ and the disparity $M$ defined in Eq. 29, will select the same set of units for treatment.*

*Proof.* The policy $D^{UT}$ treats individuals who have the highest benefit $\Delta$ in each group. The $D^{CF}$ policy treats individuals with the highest counterfactual benefit $\Delta_C$. Importantly, the policies treat the same number of individuals in the $x_0$ and $x_1$ groups. Note that, in the absence of counterfactual crossing, the relative ordering of the values of $\Delta, \Delta_C$ does not change, since

$$\Delta(u_1) > \Delta(u_2) \iff \Delta_C(u_1) > \Delta_C(u_2). \tag{86}$$

Thus, since both policies pick the same number of individuals, and the relative order of $\Delta, \Delta_C$ is the same, $D^{UT}$ and $D^{CF}$ will treat the same individuals. $\square$

## E Experiment

We apply the causal framework of outcome control to the problem of allocating mechanical ventilation in intensive care units (ICUs), which is recognized as an important task when resources are scarce [5], such as during the COVID-19 pandemic [40, 42]. An increasing amount of evidence indicates that a sex-specific bias in the process of allocating mechanical ventilation may exist [27], and thus the protected attribute $X$ will be sex ($x_0$ for females, $x_1$ for males).

To investigate this issue using the tools developed in this paper, we use the data from the MIMIC-IV dataset [17, 16] that originates from the Beth Israel Deaconess Medical Center in Boston, Massachusetts. In particular, we consider the cohort of all patients in the database admitted to the ICU. Patients who are mechanically ventilated immediately upon entering the ICU are subsequently removed. By focusing on the time window of the first 48 hours from admission to ICU, for each patient we determine the earliest time of mechanical ventilation, labeled $t_{MV}$. Since mechanical ventilation is used to stabilize the respiratory profile of patients, for each patient we determine the average oxygen saturation in the three-hour period $[t_{MV} - 3, t_{MV}]$ prior to mechanical ventilation, labeled O$_2$-pre. We also determine the oxygen saturation in the three-hour period following ventilation $[t_{MV}, t_{MV} + 3]$, labeled O$_2$-post. For controls (patient not ventilated at any point in the first 48 hours), we take the reference point as 12 hours after ICU admission, and calculate the values in three hours before and after this time. Patients' respiratory stability, which represents the outcome of interest $Y$, is measured as follows:

$$Y := \begin{cases} 0 \text{ if O}_2\text{-post} \geq 97, \\ -(\text{O}_2\text{-post} - 97)^2 \text{ otherwise.} \end{cases} \tag{87}$$

Values of oxygen saturation above 97 are considered stable, and the larger the distance from this stability value, the higher the risk for the patient. We also collect other important patient characteristics before intervention that are the key predictors of outcome, including the SOFA score [39], respiratory rate, and partial oxygen pressure (PaO$_2$). The data loading is performed using the `ricu` R-package [4], and the source code for reproducing the entire experiment can be found here.

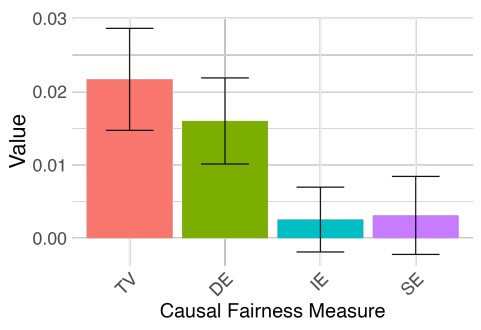

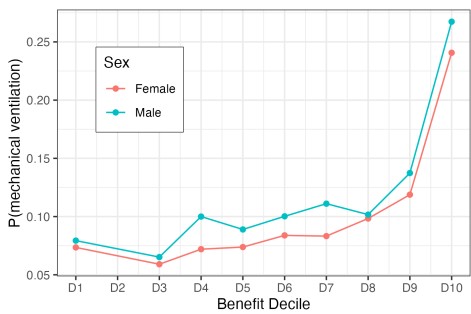

(a) Disparity in $D^{curr}$ decomposed.

(b) Benefit fairness criterion on $D^{curr}$.

Figure 6: Analyzing the existing policy $D^{curr}$.

**Step 1: Obtain the SFM.** After obtaining the data, the first step of the modeling is to obtain the standard fairness model (SFM). The SFM specification is the following:

$$X = \text{sex}, \tag{88}$$
$$Z = \text{age}, \tag{89}$$
$$W = \{\text{SOFA score, respiratory rate, PaO}_2\}, \tag{90}$$
$$D = \text{mechanical ventilation}, \tag{91}$$
$$Y = \text{respiratory stability}. \tag{92}$$

**Step 2: Analyze the current policy using Alg. 2.** Then, we perform an analysis of the currently implemented policy $D^{curr}$, by computing the disparity in resource allocation, $P(d^{curr} \mid x_1) - P(d^{curr} \mid x_0)$, and also the causal decomposition of the disparity into its direct, indirect, and spurious contributions:

$$P(d^{curr} \mid x_1) - P(d^{curr} \mid x_0) = \underbrace{1.6\%}_{\text{DE}} + \underbrace{0.3\%}_{\text{IE}} + \underbrace{0.3\%}_{\text{SE}}. \tag{93}$$

The results are shown in Fig. 6a, with vertical bars indicating 95% confidence intervals obtained using bootstrap. The decomposition demonstrates that the decision to mechanically ventilate a patient has a large direct effect of the protected attribute $X$, while the indirect and spurious effects explain a smaller portion of the disparity in resource allocation. We then compute

$$P(d^{curr} \mid \Delta = \delta, x_1) - P(d^{curr} \mid \Delta = \delta, x_0), \tag{94}$$

across the deciles of the benefit $\Delta$. In order to do so, we need to estimate the conditional potential outcomes $Y_{d_0}, Y_{d_1}$, and in particular their difference $\mathbb{E}[Y_{d_1} - Y_{d_0} \mid x, z, w]$. We fit an `xgboost` model which regresses $Y$ on $D, X, Z$, and $W$, to obtain the fit $\widehat{Y}$. The learning rate was fixed at $\eta = 0.1$, and the optimal number of rounds was chosen via 10-fold cross-validation. We then use the obtained model to generate predictions

$$\widehat{Y}_{d_1}(x, z, w), \widehat{Y}_{d_0}(x, z, w), \tag{95}$$

from which we can estimate the benefit $\Delta$. The results for the probability of treatment given a fixed decile are shown in Fig. 6b. Interestingly, at each decile, women are less likely to be mechanically ventilated, indicating a possible bias.

**Step 3: Apply Alg. 1 to obtain $D^*$.** Our next step is to introduce an optimal policy that satisfies benefit fairness. To do so, we make use of the benefit values. In our cohort of 50,827 patients, a total of 5,404 (10.6%) are mechanically ventilated. We assume that the new policy $D^*$ needs to achieve the same budget. Therefore, we bin patients according to the percentile of their estimated benefit $\Delta$. For the percentiles $[90\%, 100\%]$, all of the patients are treated. In the 89-90 percentile, only 60% of the patients can be treated. We thus make sure that

$$P(d^* \mid \Delta \in [\delta_{89\%}, \delta_{90\%}], x_1) = P(d^* \mid \Delta \in [\delta_{89\%}, \delta_{90\%}], x_0) = 0.6. \tag{96}$$

Due to the construction, the policy $D^*$ satisfies the benefit fairness criterion from Def. 2.

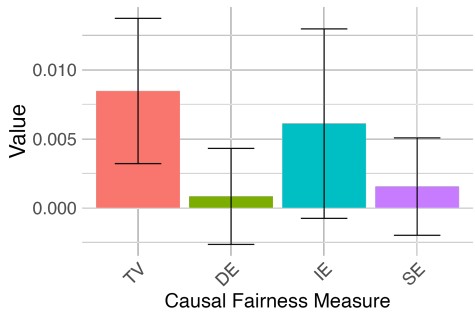
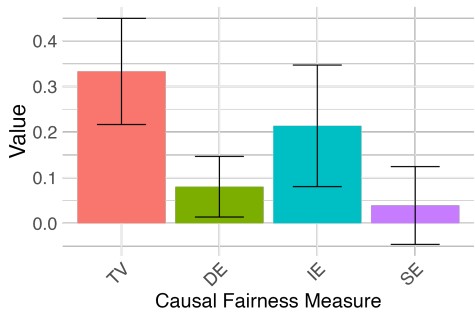

(a) Disparity in $D^*$ decomposed.  (b) Benefit $\Delta$ disparity decomposition.

Figure 7: Causal analysis of $D^*$ and $\Delta$ using Alg. 2.

**Step 4: Apply Alg. 2 to analyze $D^*$.** We next decompose the disparity of the new policy $D^*$, and also decompose the disparity in the benefit $\Delta$. We obtain the following results:

$$P(d^* \mid x_1) - P(d^* \mid x_0) = \underbrace{0.1\%}_{DE} + \underbrace{0.6\%}_{IE} + \underbrace{0.2\%}_{SE}, \tag{97}$$

$$\mathbb{E}(\Delta \mid x_1) - \mathbb{E}(\Delta \mid x_0) = \underbrace{0.08}_{DE} + \underbrace{0.21}_{IE} + \underbrace{0.04}_{SE}. \tag{98}$$

The two decompositions are also visualized in Fig. 7. Therefore, even after applying benefit fairness, some disparity between the sexes remains. The causal analysis reveals that males require more mechanical ventilation because they are more severely ill (indirect effect). They also require more mechanical ventilation since they are older (spurious effect), although this effect is not significant. Finally, males also seem to benefit more from treatment when all other variables are kept the same (direct effect, see Fig. 7b). We note that using Alg. 1 has reduced the disparity in resource allocation, with a substantial reduction of the direct effect (see Eq. 93 vs. Eq. 97).

**Step 5: Apply Alg. 3 to create $D^{CF}$.** In the final step, we wish to remove the direct effect of sex on the benefit $\Delta$. To construct the new policy $D^{CF}$ we will make use of Alg. 3. Firstly, we need to compute the counterfactual benefit values, in the world where $X = x_1$ along the direct pathway, while $W$ attains its natural value $W_{X(u)}(u)$. That is, we wish to estimate $\Delta_{x_1, W_{X(u)}}$ for all patients in the cohort. For the computation of the counterfactual values, we make use of the `xgboost` model developed above. In particular, we use the fitted model to estimate the potential outcomes

$$\widehat{Y}_{d_0, x_1, W_{X(u)}}, \widehat{Y}_{d_1, x_1, W_{X(u)}}. \tag{99}$$

The adjusted potential outcomes allow us to estimate $\Delta_{x_1, W_{X(u)}}$, after which we obtain the policy $D^{CF}$ that satisfies the CBF criterion from Def. 3.

After constructing $D^{CF}$, we have a final look at the disparity introduced by this policy. By another application of Alg. 2, we obtain that

$$P(d^{CF} \mid x_1) - P(d^{CF} \mid x_0) = \underbrace{0\%}_{DE} + \underbrace{0.5\%}_{IE} + \underbrace{0.2\%}_{SE}.$$

Therefore, we can see that the removal of the direct effect from the benefit $\Delta$ resulted in a further decrease in the overall disparity. The comparison of the causal decompositions for the original policy $D^{curr}$, optimal policy $D^*$ obtained from Alg. 1, and the causally fair policy $D^{CF}$ is shown in Fig. 8.

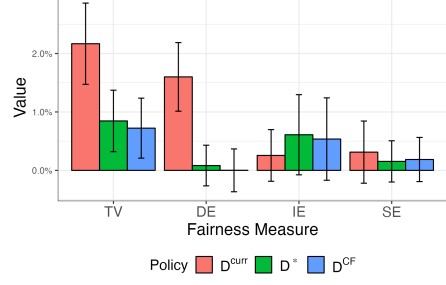

Figure 8: Causal comparison of policies $D^{curr}$, $D^*$, and $D^{CF}$.

## F    Connection of Benefit Fairness and Demographic Parity

In this appendix, we discuss the connection of the benefit fairness (BF) criterion with demographic parity (DP). The connection is given in the following formal result.

**Proposition 10** (Benefit Fairness & Demographic Parity). *Suppose that the distribution of the benefit $\Delta$ is equal between the groups $x_0, x_1$, that is*

$$\Delta \mid x_1 \overset{d}{=} \Delta \mid x_0 \tag{100}$$

*Then any decision policy $D$ satisfying benefit fairness also satisfies demographic parity.*

*Proof.*

$$P(d \mid x_1) = \sum_\delta P(d \mid x_1, \Delta = \delta)P(\Delta \mid x_1) \tag{101}$$

$$= \sum_\delta P(d \mid x_0, \Delta = \delta)P(\Delta = \delta \mid x_1) \quad \text{using BF} \tag{102}$$

$$= \sum_\delta P(d \mid x_0, \Delta = \delta)P(\Delta = \delta \mid x_0) \quad \text{using } \Delta \mid x_1 \overset{d}{=} \Delta \mid x_0 \tag{103}$$

$$= P(d \mid x_0), \tag{104}$$

implying demographic parity. $\square$

## G  Practical Considerations for Decompositions in Alg. 2

In this appendix, we discuss some practical details about how to perform the decompositions of the treatment disparity and benefit disparity as described in Alg. 2. In particular, we begin by showing that the benefit $\Delta$ can be considered as variable in the causal diagram.

**Proposition 11** (Connection of Benefit with Structural Causal Models). *Let $\mathcal{M}$ be an SCM compatible with the SFM in Fig. 1. Let the benefit $\Delta$ be defined as in Eq. 14. Let $f_Y$ be the structural mechanism of the outcome $Y$, taking $X, Z, W, D$ as inputs, together with the noise variable $U_Y$. The structural mechanism fo the benefit $\Delta$ is then given by:*

$$f_\Delta(x, z, w) = \mathbb{E}_{u_y}\left[f_Y(x, z, w, d_1, u_y)\right] - \mathbb{E}_{u_y}\left[f_Y(x, z, w, d_0, u_y)\right], \tag{105}$$

*where $\mathbb{E}_{u_y}$ integrates over the randomness in $U_Y$. Therefore, the benefit $\Delta$ is a deterministic function of $X, Z, W$, and we can add it to the causal diagram as follows:*

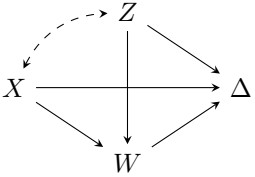

Therefore, after adding $\Delta$ to the causal diagram as in the above proposition, we can view it as another outcome variable. In particular, it also allows us to define various counterfactuals with respect to it, via the standard definitions for counterfactuals used in structural causal models. For instance, the potential outcome $\Delta_{x_0}$ (which is a random variable) is simply given by $f_\Delta(x_0, Z, W)$, where $Z, W$ are random variables but $X = x_0$ is fixed. Other, possibly nested counterfactuals are also defined analogously.

Equipped with a structural understanding of how benefit relates to the original causal model, we can now explain in more detail how the decompositions in Alg. 2 are performed. In particular, the decompositions follow the previously proposed approaches of [44, 31]:

**Proposition 12** (Computing Decompositions in Alg. 2). *Let $\mathcal{M}$ be an SCM compatible with the SFM in Fig. 1. Let the benefit $\Delta$ be defined as in Eq. 14 and suppose it is added to the causal diagram as in Prop. 11. The decompositions of the disparity in treatment allocation, and the disparity in expected*

*benefit, can be computed as follows:*

$$P(d \mid x_1) - P(d \mid x_0) = \Big( \underbrace{P(d_{x_1, W_{x_0}} \mid x_0) - P(d_{x_0} \mid x_0)}_{\text{direct effect } DE(X \to D)} \Big) \tag{106}$$

$$- \Big( \underbrace{P(d_{x_1, W_{x_0}} \mid x_0) - P(d_{x_1} \mid x_0)}_{\text{indirect effect } IE(X \to D)} \Big) \tag{107}$$

$$- \Big( \underbrace{P(d_{x_1} \mid x_0) - P(d_{x_1} \mid x_1)}_{\text{spurious effect } SE(X \leftarrow\!\dashrightarrow D)} \Big). \tag{108}$$

$$\mathbb{E}(\Delta \mid x_1) - \mathbb{E}(\Delta \mid x_0) = \Big( \underbrace{\mathbb{E}(\Delta_{x_1, W_{x_0}} \mid x_0) - \mathbb{E}(\Delta_{x_0} \mid x_0)}_{\text{direct effect } DE(X \to \Delta)} \Big) \tag{109}$$

$$- \Big( \underbrace{\mathbb{E}(\Delta_{x_1, W_{x_0}} \mid x_0) - \mathbb{E}(\Delta_{x_1} \mid x_0)}_{\text{indirect effect } IE(X \to \Delta)} \Big) \tag{110}$$

$$- \Big( \underbrace{\mathbb{E}(\Delta_{x_1} \mid x_0) - \mathbb{E}(\Delta_{x_1} \mid x_1)}_{\text{spurious effect } SE(X \leftarrow\!\dashrightarrow \Delta)} \Big). \tag{111}$$

*Furthermore, under the assumptions of the SFM in Fig. 1 both of the decompositions are identifiable and their identification expressions are given by*

$$DE(X \to F) = \sum_{z,w} \Big[ \mathbb{E}(F \mid x_1, z, w) - \mathbb{E}(F \mid x_0, z, w) \Big] P(w \mid x_0, z) P(z \mid x_0), \tag{112}$$

$$IE(X \to F) = \sum_{z,w} \mathbb{E}(F \mid x_1, z, w) \Big[ P(w \mid x_0, z) - P(w \mid x_1, z) \Big] P(z \mid x_0), \tag{113}$$

$$SE(X \leftarrow\!\dashrightarrow F) = \sum_{z} \mathbb{E}(F \mid x_1, z) \Big[ P(z \mid x_0) - P(z \mid x_1) \Big], \tag{114}$$

*where the random variable $F$ is either replaced by the treatment decision $D$ to obtain the direct, indirect, and spurious terms in Eqs. 106-108 or by the benefit $\Delta$ to obtain the terms in Eqs. 109-111.*

