# OpenReview forum: "Causal Fairness for Outcome Control"
_NeurIPS.cc/2023/Conference — NeurIPS 2023 poster_

### Official Review · Reviewer_BFhw · 2023-07-03

**Soundness:** 4 excellent
**Presentation:** 3 good
**Contribution:** 4 excellent
**Rating:** 7
**Confidence:** 4

**Summary:**

The authors analyze a new dimension of decision making: outcome control and benefit from decisions. Specifically, this work uses causal analysis to introduce new definitions of fairness on benefits and outcome control. The authors also propose new algorithms to study fairness and design fair algorithms for outcome fair decisions.

**Strengths:**

* The paper proposes new fairness notions for outcome control, that is an important aspect of many decision making settings
* The authors use a simple example to walk through all the different aspects and notions, that greatly helps the readability of the paper.
* Through the synthetic use case, the authors show when and how benefit fairness is important, and how causal path analysis could also become critical for decision making while accounting for fairer outcomes.


**Weaknesses:**

* While the appendix provides additional results on a real-world COVID dataset, the main paper has limited results on various different settings.
* While the authors assume causal knowledge, the paper does not discuss limitations explicitly for when computing counterfactuals and especially path-specific counterfactuals might become challenging in practice.
* Section 3 should have better explained which counterfactuals we compute to look at the path-specific factors and how the 3 different components are found out by decomposition.


**Questions:**

* Two works [1, 2] seem somewhat related to the authors’ notions and framework. How does the framework of decision outcome control relate to the thinking of fairness of “treatments” in [1]? Similarly, how does the notion of “benefit” the authors introduce relate to the notion of “harm” in [2]?

[1] Madras, David, et al. "Fairness through causal awareness: Learning causal latent-variable models for biased data." Proceedings of the conference on fairness, accountability, and transparency. 2019. [2] Richens, Jonathan, Rory Beard, and Daniel H. Thompson. "Counterfactual harm." Advances in Neural Information Processing Systems 35 (2022): 36350-36365.


**Limitations:**

* While the paper’s primary focus is on introducing a new theoretical framework of fairness in decision making related to outcome control and benefit, more analysis and results on different experimental settings might have been an added benefit.

---

> ### Author Rebuttal · Authors · 2023-08-09
>
> We thank the reviewer for the provided comments, which were encouraging. We provide further clarifications regarding the points raised below.
>
> ---
>
> [W1: Limited Evaluation] Please see global comment [G1].
>
> ---
>
> [W2: Computing Counterfactuals] Please see global comment [G2]. Also, we have added the following:
>
>
> - (i) We add a Proposition that states the identification property of $\Delta$. In particular, the statement is the following:
> “Under the assumptions of the extended Standard Fairness Model from Fig. 1, the benefit $\Delta(x,z,w)$ can be identified from observational data.”
> - (ii) Further we also added a comment on the fact that direct, indirect, and spurious effects of $X$ on $\Delta$ are identifiable in this settings.
> - (iii) We also remark that the $\Delta(x, z, w)$ itself may be easily adjusted to remove the direct effect of $X$. However, indirect effect counterfactuals at a covariate-specific level $x, z, w$ are more difficult to compute and may require further assumptions such as additivity or monotonicity of the response.
>
> We hope the above three changes adequately address the concern. Please let us know.
>
> [W3: Terms in the decomposition] Thanks for noting this. We have implemented this suggestion. We now spell out the expression for the direct, indirect, and spurious effects, appearing in the decomposition, to increase transparency.
>
> ---
>
> [Q1: Two related works] Thanks for pointing us to these references. You are indeed correct – the setting considered in [1] is related to ours. However, the approach taken therein differs from our approach. In particular, [1] computes three causal effects of interest: (i) total effect X -> Y; (ii) total effect D -> Y;  and (iii) total effect X -> D. Then, the paper suggests that fairness should be assessed based on these three quantities. Our framework, however, integrates the above effects more coherently, and brings forward a new definition of fairness based on first principles. We now also added a citation for [1].
>
> Regarding [2], we also found it quite interesting. However, the concept of harm / benefit defined in [2] is qualitatively different from our approach, due to the max operator used in Eq. (6) in [2]. The notion of harm / benefit in [2] considers what is in the literature known as canonical types [3] or principal strata [4]. However, our notion of benefit corresponds to the conditional average treatment effect (CATE), which measures the proportion who benefit minus the proportion of those harmed (in the language of [2]). We now clarify in the text that our notion of benefit differs from that in [2], to better contextualize our approach. Thanks for mentioning these works!
>
> [3] A. Balke and J. Pearl. Counterfactual probabilities: Computational methods, bounds and applications. In Uncertainty Proceedings 1994, pages 46–54. Elsevier, 1994
> [4] C. E. Frangakis and D. B. Rubin. Principal stratification in causal inference. Biometrics, 58(1): 21–29, 2002.

---

> > ### Comment · Reviewer_BFhw · 2023-08-10
> >
> > Thank you to the authors for addressing all my questions and doubts and adding 2 suggested fairness papers in their related work. I do think that the real-world empirical analysis is a strong case for your framework and can strengthen the theoretical work even more. This is especially true since it directly shows a tangible application to a causal method, which is generally hard to show! I do think the work is very interesting and adds a new view to fairness, so I maintain my acceptance suggestion.

---

> > > ### Author Response · Authors · 2023-08-11
> > >
> > > We again thank the reviewer for the suggestions during the review process, and also for acknowledging our answers. We are also encouraged by the fact that the reviewer sees the work as adding something novel and interesting to the literature, thank you!

---

### Official Review · Reviewer_H5h5 · 2023-07-05

**Soundness:** 3 good
**Presentation:** 3 good
**Contribution:** 3 good
**Rating:** 7
**Confidence:** 4

**Summary:**

The paper proposes a new fairness notion called benefit fairness that considers fairness in the outcomes of the decisions. In order to avoid the unidentifiability issue of the principle fairness, this paper proposes to condition on the conditional average treatment effect (CATE). The paper provides an algorithm that is proven to be optimal and satisfy benefit fairness. The paper in addition provides an algorithm to satisfy both benefit fairness and counterfactual benefit fairness.

**Strengths:**

The proposed new fairness notion is quite interesting. It is different from traditional fairness notions, yet it is well-motivated and -explained using a real-world example.

The proposed algorithms come with theoretical analysis that shows their optimality and fairness satisfaction.

A budget is involved in the problem formulation that makes the algorithms more generally applicable.


**Weaknesses:**

The connection between benefit fairness and traditional fairness notions other than principle fairness, such as demographic parity, equal opportunity, etc., is not clear. Are they consistent or do they conflict to some extent? A rigorous study may be out of the scope of this paper but some discussions will be helpful.

The measure of benefit fairness depends on the observation of the outcome Y. Can we generally assume that Y is available when we construct the decision policy? In the running example, Y is observed after a 2-year period. How can we estimate Y when we construct the decision policy before the 2-year period?

Some statements and notations are not very clear. For example, in Definition 3, $y_C$ is not defined and the definition of the pathway is not clear. My understanding is that the paper attempts to formulate path-specific counterfactual fairness, but the notations are quite confusing. In addition, $\Delta_{C}$ is not defined either. The two terms CBF and BFC may easily confuse the readers.


**Questions:**

Although I didn’t check the correctness of the proof, Theorem 2 seems quite strong. Are there any assumptions about the policy or the causal model in Theorem 2?

My understanding of benefit fairness is that, it considers the causal effect of the link X->Y. If we remove this link, then $\Delta(x_0,z,w)$ is always equal to $\Delta(x_1,z,w)$, and benefit fairness degrades to the conditional demographic parity. Thus, the meaning of benefit fairness is to make the causal effect of the link X->D match the causal effect of the link X->Y? Is this understanding correct?

If my above understanding is correct, does that mean that CBF cannot be satisfied if the link X->Y exists, because CBF generally means that  the decision should not depend on X?


**Limitations:**

None.

---

> ### Author Rebuttal · Authors · 2023-08-09
>
> We thank the reviewer for the provided comments, it was nice to see that the main strengths were appreciated! We respond point-by-point in the sequel.
>
> [W1: Connection with traditional notions] Great question, the manuscript indeed does not touch on this. We make the following observations and add the following paragraph to the text:
>
> “If the benefit distributions between groups are substantially different, a decision satisfying BF will not achieve demographic parity. However, for cases in which all causal pathways (direct, indirect, and spurious) between the protected attribute and the benefit are either zero or removed by adjustment, the Causal BF criterion implies demographic parity (i.e., equal allocation of resources by $D$)”.
>
> Furthermore, we also add a formal statement on this, which is added to the appendix due to considerations of space:
>
> “Suppose that the distribution of the benefit $\Delta$ is equal between the groups $x_0, x_1$. Then, benefit fairness implies demographic parity."
>
> The simple proof is also quite insightful for showing what is going on.
>
> Proof:
> \begin{align}
>     P(d \mid x_1) = \sum_\delta P(d \mid x_1, \Delta = \delta) P(\Delta \mid x_1)
>     \overset{\text{using BF}}{=} \sum_\delta P(d \mid x_0, \Delta = \delta) P(\Delta = \delta \mid x_1)
> \end{align}
> \begin{align}
> \overset{\Delta \mid x_1 \overset{d}{=} \Delta \mid x_0}{=} \sum_\delta P(d \mid x_0, \Delta = \delta)P(\Delta = \delta \mid x_0) = P(d \mid x_0),
> \end{align}
> implying parity. We thank the reviewer for raising this point; it indeed adds an important connection to the existing literature.
>
> [W2: Dependence on outcome Y] This is a good point. The label may indeed not be available for the cohort at hand in some settings. However, the upside is that it may be possible to use retrospective data, under the assumptions encoded in Figure 1. Therefore, one may use past data to assess the $\Delta$ quantity, possibly adjust it, and design a new and fair policy. On the other hand, if no data at all is available on the label $Y$, then designing a policy with such fairness guarantees is not possible, at least, with the level of assumptions currently considered.
>
> [W3: Notation Clarity] Thanks. We have added a clarification of what a clause $C$ is; further, we also define a pathway. The terms CBF and BFC are indeed too similar. We for that reason, throughout, use CBF for Causal Benefit Fairness, and BF for just Benefit Fairness. We hope this distinction between BF and Causal BF will be easier to follow.
>
> [Q1: Theorem 2 proof]   Good question; the main assumptions are the lack of confounding assumptions required for performing the identification of the effects necessary for constructing the policy. The second part of the answer is that the statement considers an infinite-sample, population-level case. We remark there is an additional level of difficulty for providing finite sample guarantees, which is not covered by our theorem statement. We feel these are intriguing challenges for future work.
>
> [Q2: X->D, X->Y effects] These are both great questions, thank you. There are two distinct parts of the CBF definition.
>
> (i) The first part ascertains the fairness of the $\Delta$ quantity,
> (ii) The second part ascertains that at equal levels of $\Delta$, the probability of treatment does not depend on the protected attribute.
>
> Your question seem to relate to part (i), if we understood it correctly. If there is no link X->Y, then $\Delta(x_1, z, w) = \Delta(x_0, z, w)$, which is indeed accurate. Put differently, as you noted, the absence of a direct X -> Y link implies conditional demographic parity for the $\Delta$ quantity. Note, however, that this does not imply anything about the parity for decision D, related to part (ii).
>
> Now, more generally, if the link X -> Y exists and it is considered to be discriminatory, our proposal is to first _adjust the_ $\Delta$ quantity accordingly. In other words, CBF cannot be satisfied unless we _remove the discriminatory effect from_ $\Delta$. After having this condition ascertained, benefit fairness from part (ii) can be applied to ensure a notion of equity. Thus, the fairness requirement can be thought of as a “two-step” process.   We hope this makes sense but happy to provide further elaboration.
> The note on balancing the effects X->Y and X->D is very interesting but we haven’t thought about doing that explicitly. Stll, in order to achieve the described notion of fairness, both of these pathways are affected by our procedure. Please let us know if you have any further questions, or if you see the balancing of the effects from a different perspective.

---

> > ### Comment · Reviewer_H5h5 · 2023-08-10
> >
> > Thank you for your responses. Your explanations about the CBF definition are quite helpful. I have a separate question.
> >
> > I consider the treatment benefit $\Delta$ as some sort of stratification of the population. From this perspective, how do you suggest we perform the conditioning on $\Delta=\delta$ in practice, especially when each individual has a different treatment effect? For example, do you suggest we bin the treatment effects and then condition on bins?

---

> > > ### Author Response · Authors · 2023-08-11
> > >
> > > We thank the reviewer for acknowledging our responses.
> > >
> > > Regarding $\Delta$ in practical applications: exactly, our approach would then be to use a fixed number of bins, on which we condition. The bins can correspond to quantiles of the $\Delta$ distribution, for example. The larger the sample size, the more bins we could use in practice, of course, and in the infinite sample limit this would correspond to conditioning on fixed $\Delta = \delta$ values. Actually, the approach with bins was also used in the experiment in Appendix E, where we conditioned on the percentiles of the $\Delta$ distribution (which was possible due to quite large sample size).
> > >
> > > Please let us know if there are any further questions!

---

### Official Review · Reviewer_pGG2 · 2023-07-05

**Soundness:** 3 good
**Presentation:** 2 fair
**Contribution:** 2 fair
**Rating:** 6
**Confidence:** 4

**Summary:**

This paper studies the problem where a decision maker must allocate a treatment $D$ to optimize an outcome variable $Y$ while ensuring that the decision is fair, formulating fairness as the protected attribute $X$ not having an influence on $D$. It uses a clinical decision-making process (a running example on Cancer Surgery, Fig. 2(b)) to showcase its proposed method.

The paper proposes the notion of benefit based on the potential outcomes (PO) framework. $Y_{d_0}$ denotes the outcome of a patient that didn’t undergo surgery while $Y_{d_1}$ denotes the outcome of a patient that did undergo surgery. Under PO, only one of the two potential outcomes are observed per patient. Benefit is defined as $Y_{d_1} - Y_{d_0}$. Hence, a decision maker will derive a $D^*$ that maximizes overall benefit: i.e., those patients for which $Y_{d_0}=0$ (died without surgery) and $Y_{d_1}$ (survived with surgery).

Through the running example, the paper shows how a decision maker (under imperfect information) can be unfair via $D$ despite optimizing for the benefit. It then introduces a set of algorithms to reach fair benefit.


**Strengths:**

[S1] The topic is relevant and novel within (causal) fairness. Under a heterogenous population of individuals, it is reasonable to expect that not all individuals will benefit from a treatment (captured by $D$ and its causal effect on $Y$ in the paper). The notion of benefit becomes relevant as (i) it formalizes which individuals need $D$ and which do not; (ii) it raises the potential issue of what happens when those that benefit under $D$ don’t embody societal representational fairness goals.

[S2] The use of the running example was very useful for understanding the paper’s goal.

[S3] Fig. 3 illustrates well the problem of allocating a treatment without knowing the type of patient w.r.t. to $D$.

[S4] Def. 2 (Benefit Fairness), putting aside the estimation issues, is a nice extension to other group-level fairness definitions (like Equal Opportunity). It states that individuals across the protected attribute $X$ that have the same (potential) benefit $\delta$ to gain from treatment should have the same probability of getting treated.


**Weaknesses:**

[W1] PO limitations should be discussed further: Given the use of the PO framework, there’s a lack of treatment on how the benefit ($\Delta = Y_{d_1} - Y_{d_0}$) can be estimated in a consistent and unbiased way, at least, at the individual level. For each individual, we can only observe $Y_{d_0}$ or $Y_{d_1}$: for each individual, the one observed is the factual and the other one the counterfactual. However, the paper lacks a meaningful discussion on how we can estimate the counterfactuals in the non-oracle setting, which is the one of practical interest. Even in the running example, isn’t there a risk that the clinicians are using a biased $\Delta$? If so, how would that affect the proposed algorithms? The limitations should be discussed explicitly.

In particular, the derivation and justification of Eq. (16), which is central to the algorithm(s) is not clear. Is it a statement on the same tuple $(x, z, w)$: if so, how can the same individual be benefited and harmed at the same time by the treatment? Or is it a “match” between two individuals? Also, what is $z$ in the running example (this random variable is only presented and used in Fig. 1)?

Counterfactual generation/estimation is not exploited in the example. For instance, if we have access to the SCM model, couldn’t we generate the counterfactual distribution via Pearl’s abduction, action, and prediction steps for $Y$. We could then identify the “doomed” and “safe”? This is important as Algorithm 1 can only operate meaningfully if it can identify the borders of the types of individuals (w.r.t. the notion of benefit). Line 1 in Algorithm 1 needs to be discussed further.

[W2] Stronger Section 2: This section could be tighter. For instance, what’s the role of the budget $b$ from Def. 1? Even in the oracle setting, it is possible that we can’t treat all 100 patients if the budget is low enough, no? Conversely, if the budget is high enough and $\Delta \geq 0$, we could treat all patients, no?

Further, as a follow up to W1, if there is the pair \{$Y_{d_0}$, $Y_{d_1}$\} for each individual then there is also a $\Delta$ parameter. Now, under the oracle, we can see through the future and allocate $D$ based on $\Delta$. In the non-oracle setting, there’s a brief discussion about using $W$ as a proxy that leads to unfairness: how does that translate into estimating $\Delta$? Otherwise, if we can estimate it, then why are we even considering the proxy $W$? What I’m hinting at here – do correct me if I’m wrong – is that Def. 2 needs to have some measure of uncertainty to highlight that the non-oracle allocation will have some error w.r.t. to the oracle allocation (for a fixed $b$). Otherwise, it seems like we are still in the oracle setting.

[W3] Limited evaluation: The running example is essentially the only use case. Although it shows the proposed methodology, it would’ve helped to, e.g., test the algorithms under different parameters for the same synthetic data. Similarly, could the algorithms handle other variables on top $W$ or under a second protected attribute like race? The evaluation, even for the single use case, could’ve been pushed further to show the robustness of the approach.

[W4] Some definitions are unclear or not fully explained. For instance,
In Def. 1, the role of the budget $b$ is not explained. It would’ve also helped to formulate it under the PO framework explicitly.
In Def. 2, please define $\Delta$ and $\delta$ within the definition itself. These are presented later.
In Def. 3, is it at the individual level? Under the SCM presented (Fig. 2, but also Fig. 2), how are the conditionals the same under the interventions captured by the causal pathways. Don’t we need to update for downstream effects under $\mathcal{C}$?


**Questions:**

See my questions in the Weaknesses section. I’m willing to increase my score if these questions are addressed.

**Limitations:**

Further discussion on $\Delta$ is missing. It’s a considerable limitation for the application of this definition and it’s not evaluated here theoretically.

---

> ### Author Rebuttal · Authors · 2023-08-09
>
> In the weaknesses section, the questions were good and we believe to have addressed them in a factual and robust way. Please let us know if there are any further issues!
>
> ---
>
> [W1a: PO Limitations] Thanks for pointing this out. See global comment [G1] for a detailed response.
>
> [W1b: Biased $\Delta$] This is indeed possible and is likely to be the case in practice. In some sense, using $\Delta$ is “optimal”, while clinicians probably use a less-than-perfect version of $\Delta$. We’ve included a specific remark to acknowledge this. It’s worth noting that our goal is to understand the conditions under which decision-making should be made, rather than describing how currently stakeholders make decisions – which, as you pointed out, may be suboptimal. We hope our work can serve as a normative guide to enhance decision-making in practice.
>
> [W1c: Eq. (16) justification] Thanks for the opportunity to clarify this issue, which is central to our contributions. The key distinction here is between ‘covariate-specific’ and ‘unit-level’. By ‘covariate specific’, we mean all units $u$ of the SCM that are compatible with the observed values $(x, z, w)$. The value of $\Delta$ is computed for a group of units with fixed covariates. However, for each of these units, the outcome can only be 0 or 1, as you rightly pointed out. $\Delta$ represents an average over these outcomes. Unfortunately, both ‘covariate-specific’ and ‘unit-level’ approaches are called “individual” in the literature, leading to much confusion. To address this, we have dded the following for clarity:
>
> “The values of $c, d$ are covariate-specific, and indicate the proportions of patients helped and harmed by the treatment, respectively, among all patients coinciding with the observed event $X=x, Z=z, W=w$ (i.e., all $u \in U$ s.t. $(X, Z, W)(u) = (x,z,w)$). “
>
> [W1d: What is Z?] Generally, the $Z$ can be an arbitrary set of observed confounders, whereas in our running example $Z = \emptyset$. Thanks for noting this. We have added this explicitly in the description of Figure 2.
>
> [W1e: $\Delta$ estimation] Agreed! Please see suggested improvements (i)-(iii) we mention in global answer [G1]. However, we remark that identifying doomed and safe groups would not be possible in the absence of the SCM. However, identifying the difference of the proportions $c(x,z,w) - d(x,z,w)$ is possible from observational data.
>
> ---
>
> [W2: Section 2] Thanks – both assertions are indeed correct. Firstly, we make the following addition to the text, just below Definition 1 :
>
> “The budget b is relevant for scenarios when resources are scarce, in which not all patients possibly requiring treatment can be given treatment. In such a setting, the goal is to treat patients who are most likely to benefit from the treatment, as formalized next in the text.”
>
> Further, note that if b is large enough, only the $\Delta > 0$ condition needs to be checked. This is reflected in Line 2 of Algorithm 1, and also Line 2 of Algorithm 3 (and corresponds exactly with the intuition you described). So, the importance of having a budget comes for cases when not all patients who benefit from treatment can be treated. In this case, the reasoning becomes more involved, and we need to estimate $\Delta(x,z,w)$ for all $x,z,w$ values, and treat patients with the largest benefit $\Delta$, as described in the Algorithms 1 and 3.
>
> ---
>
> [W3a: Limited Eval] Please see global response [G2].
>
> [W3b: Other Ws and second attribute] We appreciate the question, thank you. Once again, we can provide some positive answers. Firstly, the set of variables $W$ can be multidimensional (see Appendix E for a specific real-world example). Furthermore, the example in Appendix E also shows how we can have a set of confounders $Z$. Therefore, the framework is compatible with datasets of a larger dimension.
>
> Regarding the extension of the protected attribute – this is indeed possible. The simplest way of seeing this is to define the protected groups as all elements of the Cartesian product of the domains of the two attributes (e.g., multiple (sex, race) combinations). Then, for example, Eq. (15) could be an equality relating more than two different groups, but would in principle remain the same.
>
> ---
>
> [W4a: Def. Explanations] We hope the discussion on the budget provided above helped to clarify this issue. Still, let us know if you think this is sufficient. Definitions of $\Delta$ and $\delta$ are now put in Definition 2.
>
> [W4b: Individual Level Def. 3?] Thanks for asking; Yes, the first part of the definition is covariate-specific (Eq. (25)), and the second part is specific to a fixed $\delta$ level (Eq. (26)), but across covariate tuples. We now updated the text to make this transparent:
>
> “CBF requires that treatment benefit $\Delta$ should not depend on the effect of $X$ on $Y$ along the causal pathway $\mathcal{C}$, and this condition is covariate-specific, i.e., holds for any choice of covariates $x, z, w$. Additionally, the decision policy $D$ should satisfy BF, meaning that at each degree of benefit $\Delta = \delta$, the protected attribute plays no role in deciding whether the individual is treated or not. This statement is for a fixed value of $\Delta = \delta$, and possibly considers individuals with different values of $x, z, w$.“
>
> [W4c: Same conditionals?] The question pertains to Figures 4(a,b), right? (there may be a typo in the question). But you are correct – in Figure 4(a), we plot the densities of $\Delta \mid x_0$, $\Delta \mid x_1$ which correspond to Males and Females, respectively. Then, after adjusting for the indirect effect, the two densities become the same, as noted in your comment. We have now updated the Figure 4(a) label to have labels $\Delta \mid x_0$, $\Delta \mid x_1$ instead of Male, Female, to make things clearer. Furthermore, we increase the label size for Figure 4(b), so it is more clearly legible that the indirect effect is removed, i.e., the subscript is $W_{x_0}, x_1$.

---

> > ### Comment · Reviewer_pGG2 · 2023-08-10
> >
> > Dear authors, thank you for the very detailed answers. You really did go over all of my concerns. Under these proposed changes/clarifications, on top of the general comments (G1 and G2), I see a stronger paper. I will update my score accordingly.

---

> > > ### Author Response · Authors · 2023-08-11
> > >
> > > We would like to thank the reviewer for the constructive review process, in which we were able to improve our paper (specifically regarding identifiability results for counterfactuals). We also thank the reviewer for acknowledging our response, and for adjusting the grade as well.

---

### Official Review · Reviewer_9Jxk · 2023-07-10

**Soundness:** 3 good
**Presentation:** 3 good
**Contribution:** 3 good
**Rating:** 7
**Confidence:** 3

**Summary:**

The paper focuses on outcome control from a causal perspective of the decision-maker. The authors introduce benefit fairness taking the perspective of the decision maker and provide theoretical guarantee that the algorithmic result is optimal and satisfies benefit fairness. To support the decision maker in indicating potential discrimination early on, algorithm 2 evaluates the difference in benefit to demographic groups. And the paper defines causal benefit fairness with an accompanying algorithm and a guarantee of optimality.

**Strengths:**

S1 - In fairness research, the need for more causal perspectives has been a growing discussion. This research is significant in contributing to an underdeveloped area.

S2 - The research problem is well developed and simple guarantees provided with proofs in the appendix.


**Weaknesses:**

 Exposition clarity needs minor improvement. For instance, what is the relationship between the first and second paragraph of the introduction? What do the authors see as the connection between outcome control and historical biases?


**Questions:**

The authors do not actually provide a clear definition of “outcome control” as used in this paper. Based on the content, I assumed they are using the high level definition put forth by [Procedural Justice in Algorithmic Fairness, CSCW 2019], which defines outcome control as enabling the correctability and possible recourse against an individual decision. But this paper is not cited in the present paper. The authors reference [Causal Fairness Analysis, 2022], which is an ICML tutorial where section 5.3.3 contains exact wording and sentences from the present paper lines 35-38. The tutorial includes more insight clarifying that outcome control setting requires that the institution and the individual have the same utility function. It would be helpful to clarify this in the main text of the present paper. What is the definition of “outcome control” being used in this paper?


**Limitations:**

It is not clear how generalizable this framework is given the demonstration on clinical examples.

---

> ### Author Rebuttal · Authors · 2023-08-09
>
> We thank the reviewer for the time and effort in reviewing the paper. We are encouraged by the fact that the reviewer appreciated the main strengths of the paper. We provide further clarifications to the questions raised in the sequel.
>
> [W1: Historical biases] We appreciate this insightful and fundamental question, thank you. Following historical biases, certain demographic groups differ in their distribution of covariates (either confounder Z, or mediators W). The overall benefit from treatment may then _be lower in the protected group_, due to a difference in the covariates that originates historically. Examples of this are numerous (we mention in passing the example of kidney malfunction, more common in some minority groups, which in turn reduces the possible benefit of heart surgery – this example falls under outcome control). We have now added this remark to the Intro, to make the transition more smooth. We thank the reviewer for this suggestion!
>
> [Q1: Outcome Control definition] Thank you for pointing this out. We have now made clear in the Introduction that outcome control in the context of the paper refers to the specific setting described by the causal diagram in Figure 1, in which the standard fairness model is extended to include a treatment $D$ that precedes the outcome $Y$. More formally, in outcome control, the goal is to maximize a specific outcome using a known control (e.g., maximizing survival using surgery). Furthermore, it is true that this setting is often characterized by the fact that individuals and institutions have the same utility function. This is now clarified in the text. Furthermore, we also add the following explanation and citation relating to the work that you mentioned:
>
> “Therefore, in the setting of outcome control, an institution may attempt to maximize an outcome (such as survival or well-begin) using a known control (such as surgery or a welfare intervention). This is different from the concept of outcome control in the procedural justice literature [1], which refers to the settings when individuals have some influence over assigning their own outcomes.”
>
> [1] Lee, Min Kyung, et al. "Procedural justice in algorithmic fairness: Leveraging transparency and outcome control for fair algorithmic mediation." Proceedings of the ACM on Human-Computer Interaction 3.CSCW (2019): 1-26.
>
> [L1] Here, we remark that the examples of outcome control are numerous, and go beyond just the clinical setting. For instance, consider a setting of outcome control in the context of criminal justice. A judge may wish to minimize the number of individuals who recidivate, that is, minimize the outcome $Y$. The judge can make the decision $D$ to detain the individual. This example, therefore, falls under outcome control, since the judge would attempt to use the decision $D$ to minimize the outcome $Y$. The COMPAS dataset is collected from a relatively similar setting to this one. In a further example, (Coston et al., 2020) analyzed a specific setting of a child hotline. There, the intervention is to send a social welfare team (decision $D$), while the intent is to maximize welfare and minimize harm (the outcome $Y$). The main considerations of the approach, for these settings, would remain largely the same as for the clinical examples in the paper. That is, the judge would attempt to detain those individuals for whom the detention decision reduces the probability of re-offending the most; a hotline responder would intervene to calls that are most urgent, and for which there is the greatest reduction in harm; both of these are analogous to treating patients who benefit the most from treatment in a clinical setting. We reflected this discussion in the manuscript, but, please, let us know if the answer is satisfactory or further elaboration would be desirable; thank you.

---

> > ### Comment · Reviewer_9Jxk · 2023-08-16
> >
> > Thank you for the detailed response.

---

### Author Rebuttal · Authors · 2023-08-09

The authors would like to sincerely thank all the reviewers for this paper. The main strengths and novelty were clearly appreciated, and the questions raised were quite useful for us to revise and improve the paper.

In our response, we index all weaknesses with W, questions Q, and limitations L. We do not fully cite reviewer's questions (due to character limit), but try our best to provide a caption for each W/Q/L.

Here, we would like to provide two global responses, which are then cited in the individual responses as well.

---

[G1: Limited Evaluation] Unfortunately, during the submission process, the external links of the document have stopped working. In particular, the submission included the following links (the anonymized has now been shared with the Area Chair):

- (i) In a vignette that accompanies the paper, we performed inference for the running example on finite samples, but this has not been highlighted due to the missing link. Please, have a look at Supplementary Material, source-code/vignette.html. Therein, we perform the inferences described in the paper, based on the data itself (and not the SCM).
- (ii) Furthermore, in Appendix E, we performed a real-world experiment based on the MIMIC-IV dataset. This application, as the citations in the Appendix also show, comes from the literature in intensive care medicine, and is a well-known issue (sex-specific bias in allocating respirators). We hope the reviewers will find this real application compelling.

In the current version of the writing, we decided to prioritize theoretical underpinnings rather than practical demonstrations. Of course, this is a matter of taste. We thought this type of exposition would maintain the most clarity in the concepts, while we deferred the empirical studies to the supplemental materials. However, we now suggest moving the real-world experimental results from Supplement E to the main text. We are happy to hear further suggestions on how to improve the presentation from the reviewers.

---

[G2, Identification of Counterfactuals] The paper leaves the issue of identification of $\Delta$ implicit but we agree that making it explicit could add better transparency and improve the flow. We changed the manuscript to add the following points more explicitly.

- (i) We added a Proposition that states the identification property of $\Delta$. In particular, the statement is the following: “Under the assumptions of the extended Standard Fairness Model (Fig. 1), the benefit $\Delta(x,z,w)$ is identifiable from observational data.”
- (ii) We further added an Appendix with the proof of identification using the counterfactual axioms (Ch. 7, Pearl, 2000). In the Appendix, we also provide the identification expression for $\Delta(x,z, w)$.
- (iii) Below the proposition, we pointed to more general identification strategies. Interestingly, these are also applicable to cases with both observational and experimental data.
“More generally, the approach of [1] can be used for testing identifiability of $\Delta(x,z,w)$, and this approach also handles cases when both experimental and observational data are available.”
- (iv) Further we also added a comment on the fact that direct, indirect, and spurious effects of $X$ on $\Delta$ are identifiable in this settings. A proof of this claim, together with identification expressions, is now added to the appendix.
- (v) We also remark that the $\Delta(x, z, w)$ itself may be easily adjusted to remove the direct effect of $X$. However, indirect effect counterfactuals at a covariate-specific level $x, z, w$ are more difficult to compute and may require further assumptions such as additivity or monotonicity of the response.

Thus, we hope this clearly demonstrates that our approach is applicable to real data; that is, our approach does not assume the knowledge of the SCM itself.

---

### Decision · Program_Chairs · 2023-09-21

**Decision:**

Accept (poster)

**Comment:**

Novel use of causal analysis to introduce new definitions of fairness on benefits and outcome control.